# Targeted immunotherapy against distinct cancer-associated fibroblasts overcomes treatment resistance in refractory HER2+ breast tumors

Elisa I. Rivas[1,8], Jenniffer Linares[1,8], Melissa Zwick[1], Andrea Gómez-Llonin[1], Marc Guiu[2], Anna Labernadie[3], Jordi Badia-Ramentol[1], Anna Lladó[2], Lídia Bardia[2], Iván Pérez-Núñez[1], Carolina Martínez-Ciarpaglini[4], Noelia Tarazona[4], Anna Sallent-Aragay[1], Marta Garrido[1], Toni Celià-Terrassa[1], Octavio Burgués[4], Roger R. Gomis[2], Joan Albanell[1,5,6,7,9] & Alexandre Calon[1,9] ✉

About 50% of human epidermal growth factor receptor 2 (HER2)+ breast cancer patients do not benefit from HER2-targeted therapy and almost 20% of them relapse after treatment. Here, we conduct a detailed analysis of two independent cohorts of HER2+ breast cancer patients treated with trastuzumab to elucidate the mechanisms of resistance to anti-HER2 monoclonal antibodies. In addition, we develop a fully humanized immunocompetent model of HER2+ breast cancer recapitulating ex vivo the biological processes that associate with patients' response to treatment. Thanks to these two approaches, we uncover a population of TGF-beta-activated cancer-associated fibroblasts (CAF) specific from tumors resistant to therapy. The presence of this cellular subset related to previously described myofibroblastic (CAF-S1) and podoplanin+ CAF subtypes in breast cancer associates with low IL2 activity. Correspondingly, we find that stroma-targeted stimulation of IL2 pathway in unresponsive tumors restores trastuzumab anti-cancer efficiency. Overall, our study underscores the therapeutic potential of exploiting the tumor microenvironment to identify and overcome mechanisms of resistance to anti-cancer treatment.

Among the major subtypes of breast cancer (BC), HER2 over-expressing (HER2+) BC is associated with poor outcome and increased risk of relapse[1]. Anti-HER2 monoclonal antibodies (mAbs)−trastuzumab, pertuzumab- have largely improved HER2+ BC patients' prognosis by suppressing breast cancer cells (BCC) proliferation and enhancing cellular apoptosis[2]. Besides their direct cytotoxicity against HER2+ BCCs, anti-HER2 mAbs have also proven to stimulate anti-cancer immunity through antibody-dependent cell cytotoxicity (ADCC)[3,4]. Yet, treatment only benefits 40–60% of patients and disease recurrence has been reported in 15–20% of the cases due to the development of resistance mechanisms[5,6]. Recent findings indicate that HER2+ BC that became unresponsive to anti-HER2 mAbs had developed an immunosuppressive phenotype during treatment, thereby supporting the importance of stimulating anti-tumor immunity in patients treated with anti-HER2 mAbs[7].

A better understanding of solid tumors immunity has led to the development of numerous therapies aimed at stimulating the immune system against cancer cells. Nevertheless, it has become increasingly clear that only a small subset of patients are responding to immunotherapies[8,9]. For instance, the limited benefits observed to date in clinical trials do not advocate for implementing immunotherapy in current treatment of HER2+ BC patients[10,11].

It is well documented that immunotherapies benefit most to patients with immune-infiltrated tumors. In HER2+ BC patients, abundant intra-tumoral T-Lymphocytes (T cells) and Natural Killer (NK) cells associate with improved prognosis[12–15]. Conversely, immune excluded tumors unresponsive to treatment are usually displaying an enhanced mesenchymal, inflammatory and immunosuppressive tumor microenvironment (TME)[16]. A major component of the TME, the cancer-associated fibroblasts (CAFs), provides a signaling scaffold for many biological processes favoring tumor aggressiveness and poor prognosis[17,18]. For instance, CAFs are featured with cues directly enhancing cancer cells survival thereby affecting therapeutic outcome[19,20]. Also, CAFs have been associated with immunosuppression in solid tumors indicating a potential connection between the TME and resistance to ADCC[21]. Altogether, these findings suggest that the TME may strongly impact patients' response to immunomodulating therapies. However, current clinical assessment and decision-making about patients' treatment are essentially based on the evaluation of the epithelial compartment of the tumor[22,23]. Hence, to reach a more accurate and personalized management of the disease, there is an urgent need to increase our comprehension of the stromal biological processes involved in the response to treatment.

Here, we conduct a comprehensive study of HER2+ BC patients' response to anti-HER2 mAbs and identify distinct stromal traits predicting the lack of benefit from therapy. In addition, we develop a fully humanized immunocompetent ex vivo model of HER2+ BC recapitulating patients response to treatment that we apply to the discovery of therapeutic solutions to overcome the resistance to anti-HER2 mAbs.

## Results

### Stromal traits associate with resistance to HER2-targeted therapy

We first investigated whether TME attributes were associated with clinical response to anti-HER2 mAbs in HER2+ BC patients. We performed immunohistochemical (IHC) analysis in 22 cases of HER2+ BC and confirmed that tumors from patients remaining disease-free following trastuzumab treatment showed an increased number of intra-tumoral CD45 + immune cells (leukocytes) compared to tumors from relapsing patients (Fig. 1a, Supplementary Fig. 1a). Next, we performed ESTIMATE (Estimation of STromal and Immune cells in MAlignant Tumors using Expression data) analyses[24] to infer the presence of infiltrating stromal and immune cells in two independent cohorts of HER2+ BC patients treated with trastuzumab. In a first cohort of 48 patients responsive or resistant to treatment[25], immune/stroma enrichment score ratio was significantly upregulated in biopsies collected before treatment from patients achieving pathological complete response (pCR) (Fig. 1b, left panel). In a second cohort of 51 HER2+ BC cases[26], ESTIMATE analysis indicated that non-relapsing patients after adjuvant HER2-targeted treatment were also characterized by increased immune/stroma ratio (Fig. 1b, right panel).

Recent findings have brought to light the heterogeneous nature of the stroma. For instance, several functionally distinct CAFs subtypes have been identified in BC. Among them, CAF-S1 corresponds to a subset of fibroblasts specifically associated with an immunosuppressive TME[27]. Albeit being indicative of relapse in early luminal BC, CAF-S1 subtype mainly characterizes aggressive triple negative BC[27,28]. Two additional CAFs populations—pCAFs (PDPN+ CAFs) and sCAFs (S100A4+ CAFs)—have been recently described in BC[29]. While pCAFs associate with immune regulation

and worse prognosis, authors suggested that the presence of antigen-presenting sCAFs may activate the immune system thus improving clinical outcome in BC[29]. Gene set enrichment analyses (GSEA) performed in our two cohorts indicated that sCAF signature was either similarly expressed in pCR and residual disease (RD) subsets or significantly enriched in relapse-free patients compared to relapsing ones (Fig. 1c). In contrast, we found that CAF-S1 (Fig. 1d) and pCAF expression signatures (Fig. 1e) were both significantly enriched in HER2+ BC patients unresponsive to anti-HER2 mAbs and prone to relapse after treatment. Therefore, the immunosuppressive functions related to CAF-S1 and pCAF may associate with resistance to trastuzumab in HER2+ BC patients.

### Fully humanized ex vivo model of breast cancer recapitulates immune exclusion and resistance to anti-HER2 mAbs

Prompt by these observations, we aimed to visualize the interaction between HER2+ BCCs, immune cells and CAFs upon anti-HER2 treatment. However, few suitable preclinical models have been developed to date. Because therapeutic human mAbs do not necessarily cross-react with animal tissues, in vivo experiments are generally performed using surrogate antibodies that differ from the ones being developed for patients, thus restricting any extrapolation to the clinical setting. Alternatively, humanized in vitro and in vivo models are often lacking key stromal components or may not be adapted to extensive and large scale therapeutic evaluation[30,31].

To overcome this obstacle, we have developed an original fluorescent-coupled immunocompetent ex vivo model of human HER2+ BC tumors (HER2+ 3DiBC) allowing functional analyses in three dimensions (3D). Importantly, our model enables measuring the tumor response to therapies through cancer cells specific bioluminescence in a high throughput fashion.

HER2+ BCCs (Supplementary Fig. 1b) were genetically engineered to express mCherry and the luciferase enzyme, which allow the tracking of BCCs by fluorescence (Fig. 2a) or by bioluminescence quantification as readout of BCCs abundance (Supplementary Fig. 1c). Breast fibroblasts (BFs) isolated from HER2+ BC patients and GFP labeled were co-cultured with BCCs in basement membrane-like matrix drops. Development of BCCs spheroids and BFs networks in 3D was observed after 4 days co-culture (Fig. 2a). Bioluminescence quantification showed that BCCs proliferation was enhanced in co-culture and positively correlated with the amount of BFs (Fig. 2b; Supplementary Fig. 1d). Following BCCs expansion and BFs colonization, naïve primary immune cells (ICs) collected from the peripheral blood of healthy donors were inoculated into the 3D system. We did not observe intrinsic immune activity against BCCs within the usual 3 days experimental setting. Rather, naïve ICs were able to enhance BCCs expansion both in presence (Fig. 2c; Supplementary Fig. 1e) and in absence of BFs (Supplementary Fig. 1f).

The HER2+ 3DiBC model allowed us to investigate the impact of stromal cells on the efficacy of HER2-targeted treatment. Trastuzumab administration resulted in a 16% reduction of BCCs when cultured alone (Fig. 2d, Supplementary Fig. 1g, h) and of 9% BCCs reduction when co-cultured with BFs (Fig. 2d; Supplementary Fig. 1h). However, in presence of ICs, HER2-targeted treatment induced a profound decrease (-39%) of BCCs (Fig. 2d; Supplementary Fig. 1g, h) associated with abundant immune infiltration/expansion (Fig. 2e, f) in absence of BFs. Remarkably, the introduction of BFs instigated a robust decrease of intra-tumoral immune infiltration/expansion (Fig. 2e, g) and abolished trastuzumab-induced ADCC against BCCs (Fig. 2d; Supplementary Fig. 1h). Furthermore, increasing abundance of BFs in HER2+ 3DiBC reduced trastuzumab-induced ADCC in a dose dependent manner (Supplementary Fig. 1i). These data demonstrate that HER2+ 3DiBC model recapitulates ex vivo a mechanism of resistance observed in BC patients.

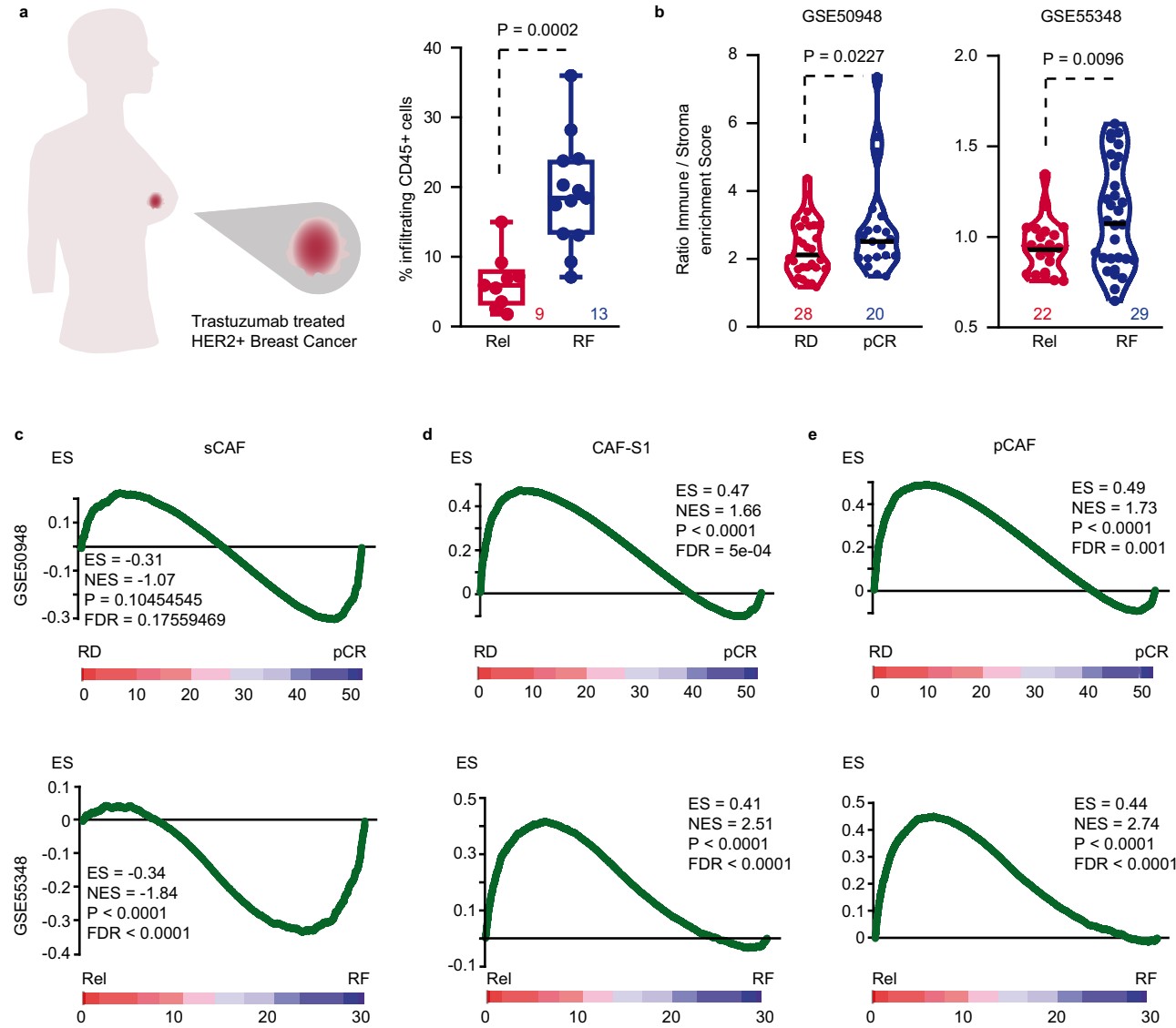

**Fig. 1 | Stromal traits define resistance to trastuzumab. a** Percentage of CD45+ cells in HER2+ BC tumors comparing relapse-free (blue) to relapsing patients (red) after trastuzumab-based therapy. Central mark indicates the median, box extends from the 25 to 75th percentiles, whiskers represent the maximum and minimum data point. Number of patients in each group is indicated. **b** Violin plots of immune/stroma enrichment score ratio for patients with pCR (blue) vs. RD (red) (GSE50948) and for relapsing (red) vs. relapse-free patients (blue) (GSE55348). Number of patients in each group is indicated. **c**−**e** Gene-set enrichment analyses (GSEA) in tumor samples collected before treatment of (**c**) sCAF, (**d**) CAF-S1, and (**e**) pCAF gene expression signatures comparing pCR vs. RD (upper row) and relapsing vs. relapse-free (lower row) HER2+ BC after trastuzumab treatment. Rel relapsing patients, RF relapse-free patients, pCR pathological complete response, RD residual disease, ES enrichment score, NES normalized enrichment score, FDR false discovery rate. Two-sided, unpaired *t* test *p* values (P) are indicated for (**a**, **b**). GSEA nominal *p* value (P) and FDR-adjusted *p* value are indicated for (**c**−**e**). Source data are provided as a Source Data file.

## IL2 activity correlates with better prognosis in trastuzumab treated patients

Immune exclusion was evident in tumors from patients unresponsive to anti-HER2 mAbs as well as in our HER2+ 3DiBC model. We thus sought to identify the immune gene expression programs associated with the response to anti-HER2 mAbs. To this end, we used ~4500 gene sets as surrogates representing cell types, states, and perturbations within the immune system, generated by manual curation or available in the Broad Institute database[32].

Among them, the gene expression program induced by interleukin 2 (IL2) in NK cells (NK-IL2RS; NK-IL2 response signature) was inversely correlated with CAF-S1 and pCAF expression in GSE55348 and with pCAF in GSE50948 (Supplementary Fig. 2a). Remarkably, NK-IL2RS was one of the top signatures upregulated before treatment in BC patients that responded to anti-HER2 mAbs. In the first cohort, NK-IL2RS was strongly upregulated in patients achieving pCR compared to those showing evidence of RD following treatment (Fig. 3a). In the second cohort, GSEA indicated that NK-IL2RS was elevated in patients that remained free of relapse after therapy (Fig. 3b). Consequently, low NK-IL2RS expression was an effective predictor of recurring disease after treatment (Fig. 3c), independently of the main clinical variables (Fig. 3d). IHC analyses in our 22 cases of HER2+ BC confirmed that tumors from patients that remained disease-free following anti-HER2 mAbs treatment showed an increased number of intra-tumoral CD56+ immune cells (NK cells) compared to tumors from relapsing patients (Fig. 3e, Supplementary Fig. 2b). Thus, our findings underscore that stromal characteristics—increased CAF-S1 and pCAF, decreased NK-IL2RS—associate with resistance to anti-HER2 mAbs. These cues are conserved across unresponsive patients and may be of use as actionable biomarkers for combinatorial therapy.

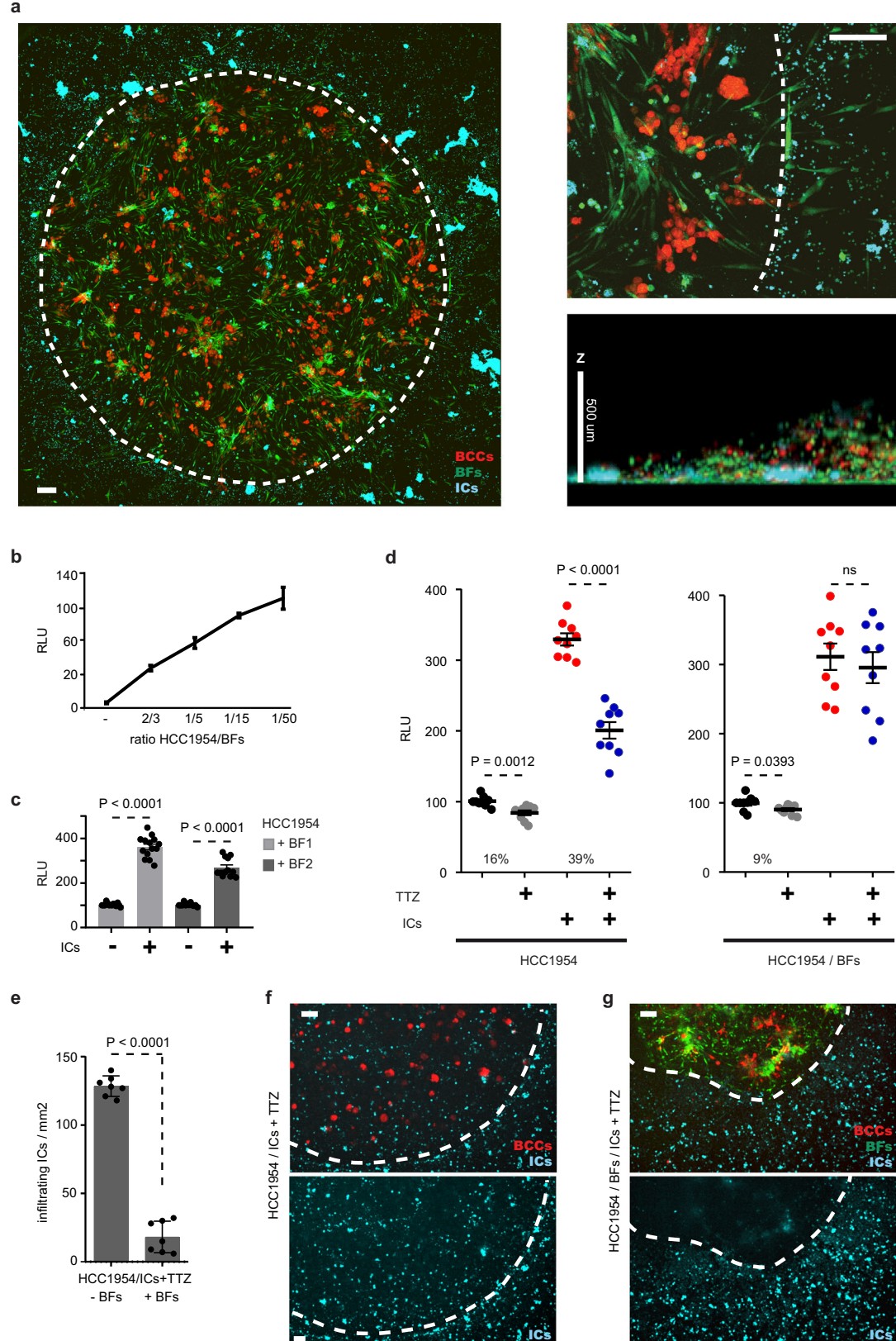

## IL2 restores immune infiltration/expansion and anti-HER2 mAbs dependent anti-cancer immunity

To further dissect the contribution of IL2 to anti-HER2 mAbs-induced ADCC, we took advantage of our HER2+ 3DiBC preclinical tumor model that we treated with trastuzumab and recombinant human (rh) IL2. Of note, IL2 receptor B (*IL2RB*) expression was only

detectable in ICs compared to BC cells and BFs (Supplementary Fig. 2c), which implies that ICs are the main effector cells for IL2 in our model. Trastuzumab and rhIL2 combination strongly enhanced immune infiltration/expansion leading to an 80% reduction of BCCs without significant effect over BFs (Fig. 3f), which coincides with the association between IL2 activity in NK cells and

**Fig. 2 | HER2+ 3DiBC recapitulates immune exclusion and resistance to tras-tuzumab. a** Confocal bioimaging of HER2+ 3DiBC tumor (left panel). Magnification of inside/outside 3D tumor co-culture interface illustrates immune exclusion from the tumor (right upper panel). Dashed lines delineate tumor inner and outer compartments. Representative of $n = 3$ HER2+ 3DiBCs. Scale bars: 50 μm. 3D tumor Z-axis is indicated (right lower panel). Red: breast cancer cells (BCCs; HCC1954); Green: breast fibroblasts (BFs); Cyan: immune cells (ICs). **b** Bioluminescent tracking of HCC1954 in HER2+ 3DiBCs with increasing amounts of BFs. $n = 3$ biologically independent experiments. Values are mean ± s.d. **c** Bioluminescent tracking of HCC1954 in HER2+ 3DiBCs with or without ICs. $n = 14$ (HCC1954/BF1; light gray) and $n = 12$ (HCC1954/BF2; dark gray) HER2+ 3DiBCs examined per condition, from 7 biologically independent experiments. Values are mean ± s.e.m. **d** Bioluminescent tracking of HCC1954 in HER2+ 3DiBCs with ICs (red), TTZ (gray) or ICs + TTZ (blue) in absence (left panel) or in presence of BFs (right panel; HCC1954/BFs in a 1/5 ratio). IgG1 was used as control for TTZ. $n = 9$ HER2+ 3DiBCs examined per condition, from 3 biologically independent experiments. Values are mean ± s.e.m. **e** ICs abundance in TTZ-treated HER2+ 3DiBCs in presence or absence of BFs. $n = 7$ biologically independent experiments. Values are mean ± s.d. **f, g** Representative bioimaging of treated HER2+ 3DiBC from (**e**). Scale bars: 50 μm. **f** Micrograph of ICs infiltrating the tumor in absence of BFs representative of $n = 3$ independent samples. **g** Micrograph of immune exclusion in presence of BFs representative of $n = 3$ independent samples. Dashed lines delineate tumor inner and outer compartments in (**f, g**). Red breast cancer cells (BCCs; HCC1954), Green breast fibroblasts (BFs), Cyan immune cells (ICs). TTZ trastuzumab, RLU relative luminescence units. Two-sided, unpaired $t$ test $p$ values (P) are indicated for (**b**–**e**); ns indicates non-significant. Source data are provided as a Source Data file.

trastuzumab effectiveness observed in patients. In this experimental setting, immune infiltration/expansion was rapidly detected, reaching its maximum level after 12 to 24 h treatment compared to the low abundance of immune cells observed upon trastuzumab monotherapy (Fig. 3g, h; Supplementary Fig. 2d). As a result, ICs invasion upon dual treatment correlated with a cumulative reduction of BCCs over time (Fig. 3g). These findings are uncovering the potential of IL2 to overcome resistance to trastuzumab and demonstrate the translational relevance of our ex vivo model for probing personalized treatments for cancer patients.

## FAP defines HER2+ BC patients with low IL2 activity and high CAF-S1/pCAF content

Recombinant IL2 (Aldesleukin) is a powerful immunomodulator used to treat patients with metastatic renal cell carcinoma and malignant melanoma. However, severe adverse reactions are associated with systemic IL2 treatment, therefore hampering its therapeutic application[33,34]. Consequently, an original immunocytokine composed of IL2v -a variant with preferential affinity for T and NK cells- fused to a FAP-targeting antibody (FAP-IL2v; Simlukafusp Alfa) was designed to direct and retain IL2 activity into FAP-expressing TME while reducing IL2 adverse systemic effects[35]. FAP-IL2v is currently being evaluated in a phase I clinical trial in combination with trastuzumab (NCT02627274).

Interestingly, FAP was recently identified as a marker specifically expressed by BC CAF-S1[27] and pCAFs (Supplementary Data 1). Since the CAF-S1 and pCAF subsets characterize HER2+ BC unresponsive to trastuzumab (Fig. 1d, e), we conducted a detailed analysis of FAP expression pattern in HER2+ BC. IHC analysis performed in BC patients showed that increased FAP protein expression was detected in the TME of tumors unresponsive to trastuzumab (Fig. 4a, b). In vitro, *FAP* mRNA was exclusively expressed by BFs compared to immune and epithelial cells (Supplementary Fig. 3a). Furthermore, *FAP* level robustly correlated with CAF-S1 or pCAF expression and inversely correlated with NK-IL2RS expression in HER2+ BC patients (Fig. 4c).

Collectively, our findings indicate that HER2+ BC tumors resistant to anti-HER2 mAbs display upregulated CAF-S1/pCAF/FAP expression as well as reduced immune cells abundance and low IL2 activity. Therefore, the capacity of FAP-IL2v to increase IL2 bioavailability into a FAP-enriched stroma may strongly benefit those patients unresponsive to trastuzumab.

## FAP-IL2v enhances ADCC in HER2+ human BC assembled ex vivo

To test this hypothesis, we sought to functionally dissect FAP-IL2v ability to modulate trastuzumab efficiency against BCCs in our ex vivo setting (Fig. 4d). We first confirmed FAP protein expression by BFs in HER2+ 3DiBC (Fig. 4e, Supplementary Fig. 3b). Next, HER2+ 3DiBC tumors were pretreated with FAP-IL2v or untargeted DP47-IL2v control. Unbound compounds were retrieved after incubation to avoid interactions with free FAP/DP47-IL2v. In the absence of ICs, trastuzumab and FAP-IL2v combined therapy did not induce significant

cytotoxicity against BCCs when compared to HER2+ 3DiBC tumors inoculated with trastuzumab and DP47-IL2v (Fig. 4f, Supplementary Fig. 3c). In the presence of ICs, trastuzumab/FAP-IL2v regimen neither impacted BCCs abundance in a model lacking FAP-expressing BFs when compared to trastuzumab/DP47-IL2v or trastuzumab monotherapy (Supplementary Fig. 3d). Indeed, BFs were specifically recognized by the FAP-targeting antibody fragment of FAP-IL2v (Supplementary Fig. 3e). Also, only a mild anti-cancer response was observed in presence of ICs and BFs upon FAP-IL2v monotherapy compared to DP47-IL2v (Fig. 4g, Supplementary Fig. 3f). However, dual treatment with FAP-IL2v and trastuzumab resulted in abundant intratumoral infiltration/expansion of immune cells (Fig. 4h; Supplementary Fig. 3g, h) associated with a strong reduction of BCCs (Fig. 4g; Supplementary Fig. 3f, g). Of note, despite the cytotoxic effect imposed on BCCs by FAP-IL2v and trastuzumab regimen in presence of ICs, BFs remained unaffected by the treatment (Supplementary Fig. 3g).

## FAP-IL2v enhances ADCC through NK cells activation

To further characterize the immune reaction occurring upon treatment, we assessed the presence of immune biomarkers in HER2+ 3DiBC whole tumors. qRT-PCR analyses indicated that highest *CD45* mRNA level (leukocytes) was reached upon combination of trastuzumab with FAP-IL2v (Fig. 5a), thus correlating with the increased abundance of immune cells observed in this setting. Remarkably, this increase was accompanied by a dramatic upregulation of the NK cells marker *NCR1* (Fig. 5a). In addition, we detected elevated mRNA level of interferon gamma (*IFNG*)−a cytokine produced predominantly by T and NK cells−upon trastuzumab and FAP-IL2v regimen (Fig. 5b; Supplementary Fig. 4a). Also, mRNA levels of Perforin 1 (*PRF1*) and Granzyme B (*GZMB*)−two key effectors of ADCC[36]−were upregulated in this condition (Fig. 5b; Supplementary Fig. 4a).

The whole immune cells preparation used in HER2+ 3DiBC contains different cell subtypes that may impact anti-cancer response. Yet to take place, anti-cancer immunity needs immune cells to infiltrate into the tumor. We thus aimed to establish the cellular composition of the immune infiltrate that associates with response to treatment. To this end, BFs, BCCs, and CD45+ ICs were purified by FACS from HER2+ 3DiBC tumors (Supplementary Fig. 4b). qRT-PCR analyses on sorted ICs confirmed the upregulation of *PRF1*, *GZMB* and *IFNG* upon trastuzumab and FAP-IL2v combined therapy compared to controls (Supplementary Fig. 4c, d). The immune infiltrate was further separated by flow cytometry into T cells CD4+ and CD8+ subtypes, CD56+ NK cells expressing CD16−the Fc receptor mediating ADCC[36]−, B cells and monocytes as detailed in Supplementary Fig. 4e. Analysis of immune subpopulations showed no significant increase of CD4+, CD8+ T cells or B cells abundance upon treatment and in some cases, a decrease of CD14+ monocytes (Supplementary Fig. 4f). However, CD56+ NK, CD56+/CD16+ NK and CD3+/CD56+ natural killer-like T (NKT) cells abundance was consistently increased within the immune infiltrate

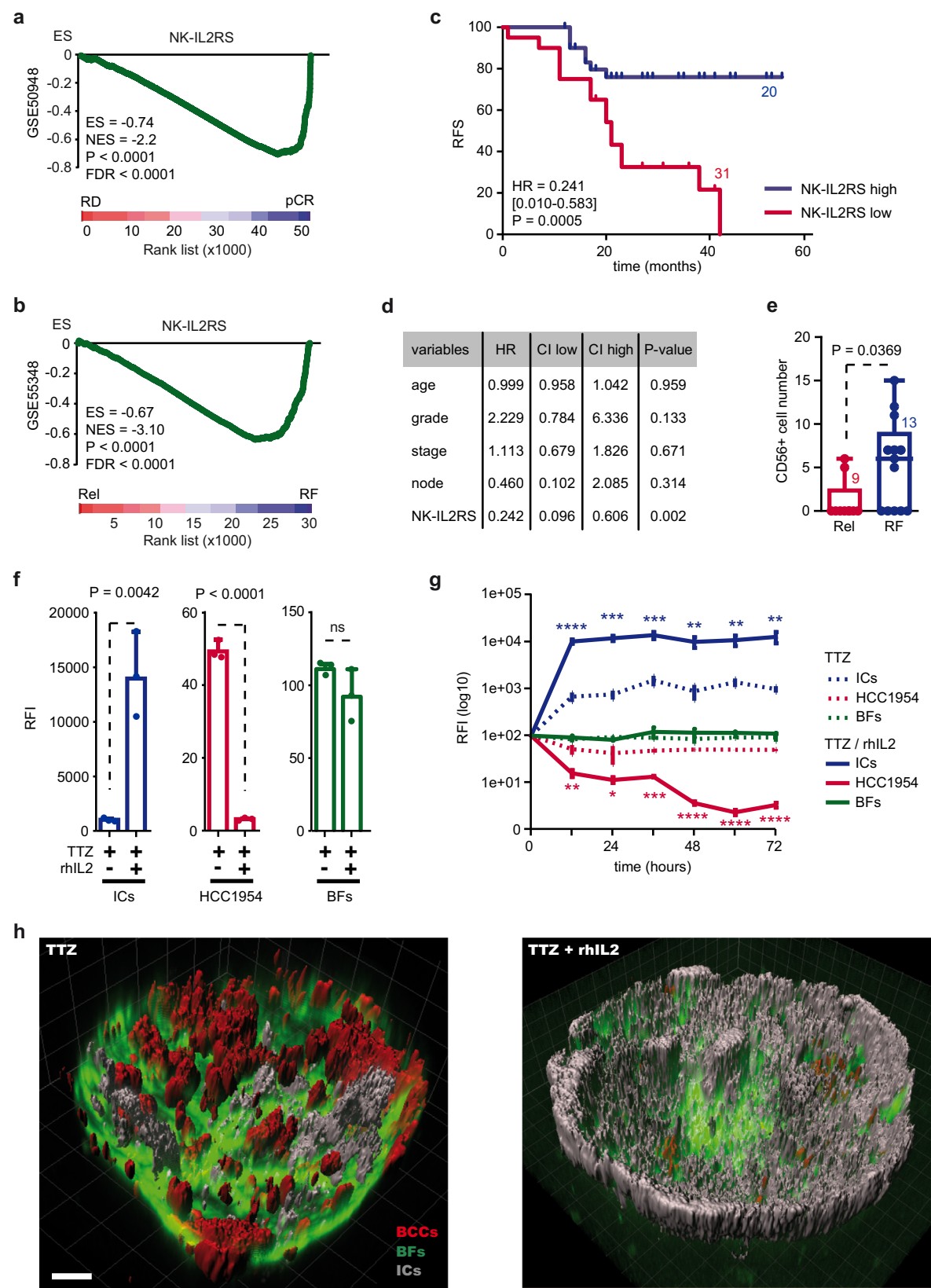

from trastuzumab plus FAP-IL2v treated HER2+ 3DiBC compared to controls (Fig. 5c, d; Supplementary Fig. 5a).

This finding prompted us to investigate the relevance of NK and NKT cells in HER2+ BC treated patients. To this end, we used NK and NKT cell signatures identified by single-cell RNA sequencing as surrogate markers[37]. In this setting, NK and NKT cells were

characterized by high levels of *CD56* and *CD16* levels, with *CD3* markers specifically upregulated in NKT cells compared to NK cells (Supplementary Fig. 5b). GSEA in aforementioned cohorts indicated that NK and NKT gene signatures were both strongly upregulated in patients achieving pCR and in patients remaining relapse-free after treatment (Fig. 5e), which confirmed that these

**Fig. 3 | Response to IL2 enhances anti-cancer immunity. a, b** GSEA of NK-IL2RS gene set comparing (**a**) pCR vs. RD and (**b**) relapsing vs. relapse-free HER2+ BC after TTZ treatment. **c** Kaplan–Meier curve shows relapse-free survival (RFS) of HER2+ BC patients treated with TTZ presenting low (red) or high (blue) expression levels of NK-IL2RS. Hazard ratio (HR) and log-rank test *p* value (P) are indicated. Number of patients in each group is indicated. **d** Multivariate Cox regression model analysis and *p* values of NK-IL2RS, adjusted by stage, grade, age and lymph nodes positivity. **e** CD56+ cell number in HER2+ BC tumor cores comparing relapse-free (blue) to relapsing patients (red) after TTZ-based therapy. Central mark indicates the median, box extends from the 25 to 75th percentiles, whiskers represent the maximum and minimum data point. Number of patients in each group is indicated. **f** Cell-specific relative fluorescence intensity (RFI) of ICs (blue), HCC1954 (red) and BFs (green) in HER2+ 3DiBCs after 72 h treatment with TTZ or TTZ + rhIL2. *n* = 3 biologically independent experiments. Values are mean ± s.d. **g** 72 h follow-up of cell-specific RFI from (**f**). TTZ-treated HER2+ 3DiBCs show increased abundance of ICs after 12/24 h (blue line), decreased presence of HCC1954 over time (red line) in presence of rhIL2. *n* = 3 biologically independent experiments. Values are mean ± s.d. **h** Micrographs of HER2+ 3DiBC IMARIS analysis from (**f**). Left panel: immune excluded tumor in absence of rhIL2. Right panel: Immune cells abundance is associated with a strong reduction of epithelial cancer cells in presence of rhIL2. Red breast cancer cells (BCCs; HCC1954), Green breast fibroblasts (BFs), Gray immune cells (ICs). Scale bar: 100 µm. Rel relapsing patients, RF relapse-free patients, pCR pathological complete response, RD residual disease, ES enrichment score, NES normalized enrichment score, FDR false discovery rate, TTZ trastuzumab. GSEA nominal *p* value (P) and FDR-adjusted *p* value are indicated for (**a, b**). Two-sided log-rank test (**c, d**) and two-sided, unpaired *t* test (**e**–**g**) *p* values (P) are indicated; ***P < 0.001; **P < 0.01; *P < 0.05; ns non-significant. Source data and exact *p* values for (**g**) are provided as a Source Data file.

cell subsets associate with treatment efficiency in the clinical setting.

Interestingly, athymic nude mice are maintaining robust NK cells activity while being deficient for functional T cells[38]. We took advantage of this model to dissect the contribution in vivo of NK cells to the action of FAP-IL2v. HCC1954 cells were implanted subcutaneously into nude mice. After cancer cells inoculation, fibroblasts of murine origin were recruited by the nascent tumor. Consequently, macroscopic tumors expressed stromal FAP protein at the time of treatment initiation (Supplementary Fig. 5c). Trastuzumab or FAP-IL2v administered alone had a very modest therapeutic effect (Fig. 5f). However, combined treatment with trastuzumab and FAP-IL2v induced a pronounced anti-cancer response compared to control regimens (Fig. 5f), thus suggesting a prominent role of NK cells in driving trastuzumab/FAP-IL2v treatment efficiency. We tested whether in vivo observations may be replicated in our ex vivo BC model. To do so, untouched NK cells were purified from freshly collected PBMCs and added to HER2+ 3DiBC. Here again, trastuzumab and FAP-IL2v combination induced a significant reduction of BCCs compared to controls (Supplementary Fig. 5d), which could not be achieved in the absence of BFs (Supplementary Fig. 5e). Of note, therapeutic effectiveness was abrogated upon administration of anti-CD16 blocking antibody (Fig. 5g), thus further underscoring a trastuzumab/FAP-IL2v-dependent ADCC mechanism. Overall, our data underscore the pertinence of HER2+ 3DiBC in recapitulating crucial biological processes associated with BC patient response to treatment and indicate that FAP-IL2v may restore trastuzumab-mediated ADCC against BCCs in unresponsive HER2+ BC.

## FAP is a biomarker of TGF-beta activity in HER2+ BC microenvironment

FAP was originally identified as a TGF-beta target in CAFs from colorectal cancer where its upregulation predicted shorter disease-free intervals[39]. In BC, as mentioned previously, FAP is a marker of immunosuppressive CAF-S1[27] and pCAF (Supplementary Data 1) subsets. In addition, CAF-S1 subpopulation accumulates in metastatic lymph nodes where it enhances BCCs migration and EMT initiation in a TGF-beta-dependent manner[40]. However, it remains unclear whether CAF-S1/pCAF phenotypes and FAP expression are depending on TGF-beta activity in BC stroma. Hence, we analyzed the TGF-beta pathway activation in our patient cohorts. To this end, we used stromal TGF-beta response signatures (TBRS) derived from fibroblasts, macrophages, endothelial, NK and T cells (F-TBRS, Ma-TBRS, endo-TBRS, NK-TBRS and T-TBRS respectively)[20]. We found that the expressions of all five TBRS (as well as *TGFB* levels in the GSE55348 dataset) were tightly correlated with *FAP* expression (Fig. 6a; Supplementary Fig. 6a). Corroborating these observations, western-blot analysis of rhTGF-beta1-treated BFs showed that FAP protein expression depended directly on TGF-beta pathway activation (Fig. 6b). Similar to CAF-S1 and pCAF, F-TBRS expression was upregulated in patients with RD (Fig. 6c) and in patients relapsing after treatment (Fig. 6d). With the exception of Endo-TBRS that was increased in pCR tumors from GSE50948, Endo-TBRS in GSE55348 as well as Ma-TBRS, NK-TBRS and T-TBRS in both GSE50948 and GSE55348 did not associate with clinical response to therapy (Supplementary Fig. 6b). Next, we explored the potential association between the TBRSs and the CAF-S1 or pCAF subsets. Transcriptomic analyses revealed that F-TBRS was strongly correlated with CAF-S1 and pCAF gene set expression in the two patient cohorts (Supplementary Fig. 6c). We obtained equivalent results overall using Ma-TBRS, endo-TBRS, NK-TBRS and T-TBRS (Supplementary Fig. 6d). Altogether, these findings indicate that the increased stromal TGF-beta activity may participate to BFs specification into FAP+ CAF-S1/pCAF in HER2+ BC.

## FAP-IL2v efficiency depends on TGF-beta activated tumor microenvironment

The above data suggest that the activation of TGF-beta pathway in BFs decreases trastuzumab effectiveness against BCCs. Yet, TGF-beta 1 also increases FAP protein expression by BFs, thus providing a potential actionable biomarker in resistant tumors. We sought to investigate whether FAP-IL2v efficiency was depending on TGF-beta pathway activation. First, we measured the levels of *TGFB1* and assessed TGF-beta activity in HER2+ 3DiBC. We observed that the expression of *TGFB1* was superior in BFs compared to BCCs (Supplementary Fig. 6e). However, TGF-beta pathway activation was significantly increased in BFs when co-cultured with BCCs (Supplementary Fig. 6f). In addition, TGF-beta pathway activation positively correlated with BCCs/BFs ratio and was completely abolished upon treatment with TGF-beta pathway inhibitor (Fig. 6e, Supplementary Fig. 6f). This suggests that additional mechanisms needed for TGF-beta activity in BFs (e.g., TGF-beta ligand activation[41]) may require BCCs. Corroborating these observations, western-blot analysis of BFs purified from HER2+ 3DiBC confirmed that FAP expression depended directly on BFs/BCCs interaction and TGF-beta pathway activation (Fig. 6f). Conversely, binding of the anti-FAP mAb fragment of FAP-IL2v to BFs was abrogated upon treatment with TGF-beta pathway inhibitor (Fig. 6g, h).

We then compared the FAP-IL2v efficacy in low and high TGF-beta activated environment. To this end, HER2+ 3DiBC tumors were pretreated with TGF-beta pathway inhibitor in order to recapitulate a TME with low TGF-beta activity. To ensure that TGF-beta pathway inhibition only affected BFs, we used BCCs unresponsive to TGF-beta (Supplementary Fig. 6g). Also, the TGF-beta pathway inhibitor was retrieved prior to FAP-IL2v/trastuzumab treatment and ICs inoculation. After 3 days of treatment, we observed that FAP-IL2v was able to restore trastuzumab activity specifically in tumors with a TGF-beta activated microenvironment (Fig. 6i). Overall, our findings suggest that FAP-IL2v capacity to enhance trastuzumab-mediated ADCC in BC tumors depends on a pre-established TGF-beta rich microenvironment.

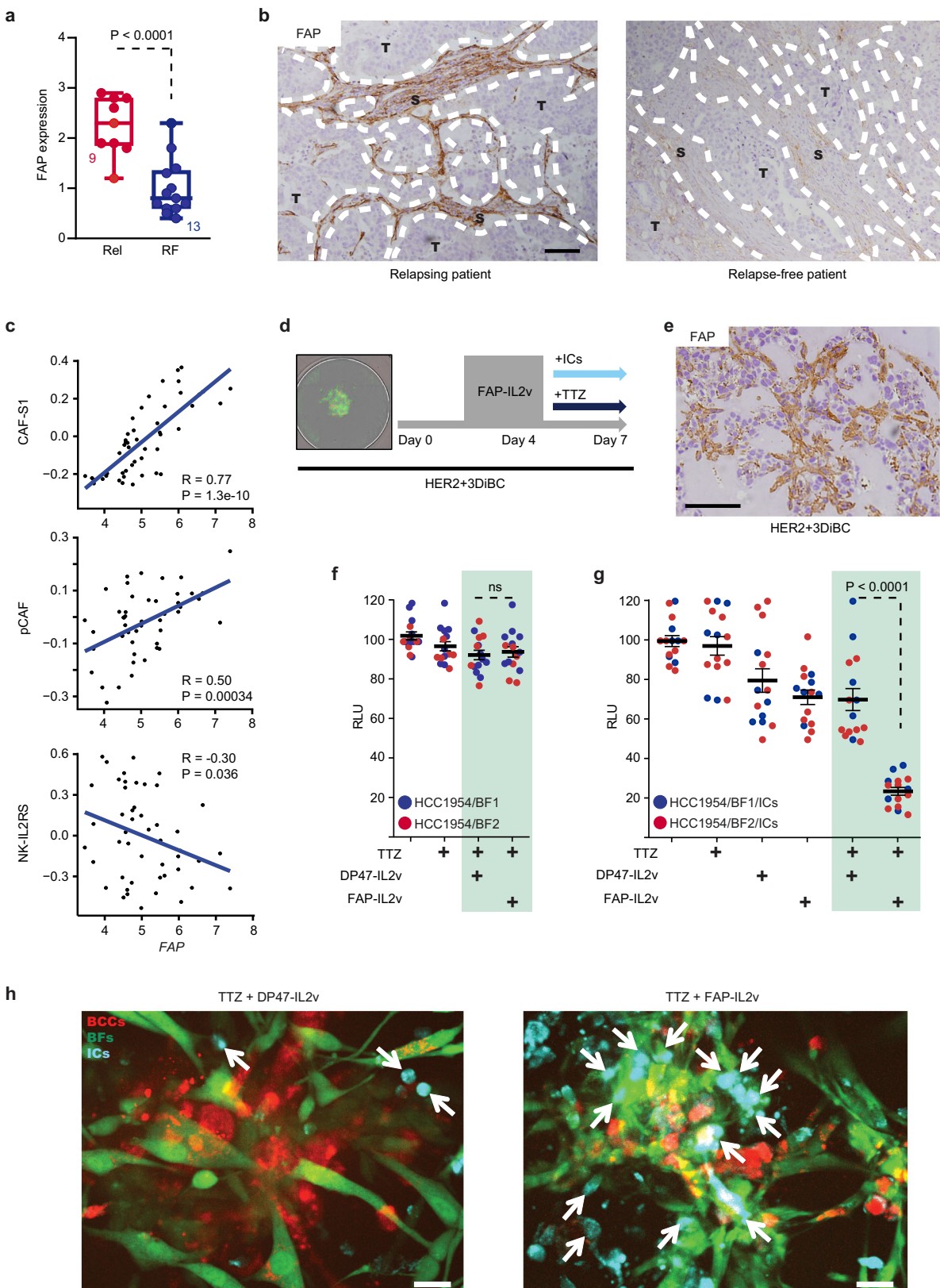

## Discussion

Advances in the field of immuno-oncology are offering great opportunities to improve the management and treatment of patients with cancers of distinct origin. For instance, the fact that mAbs-based targeted therapies are able to locally activate immune mechanisms has led to the speculation that ADCC might be in great part responsible for their effectiveness against cancer cells. Along this line, our observations in patients indicate that HER2+ breast tumors with increased expression of NK-IL2RS, NK and NKT cell signatures previous to treatment will respond better to anti-HER2 mAbs-based therapy. Conversely, our analyses reveal that the resistance to trastuzumab in HER2+ BC patients is largely contributed by tumor-associated non-

**Fig. 4 | FAP-IL2v restores anti-cancer immunity. a** FAP expression in HER2+ BC tumors comparing relapse-free (blue) to relapsing patients (red) after TTZ-based therapy. Central mark indicates the median, box extends from the 25 to 75th percentiles, whiskers represent the maximum and minimum data point. Number of patients in each group is indicated. **b** Micrographs of FAP expression in HER2+ BC human tumor samples from (**a**). T tumor, S stroma. Scale bar: 100 μm. Representative of *n* = 22 tumors. **c** Correlation between *FAP* expression levels and CAF-S1, pCAF or NK-IL2RS in HER2+ BC patients (*n* = 48) from GSE50948 dataset. Correlation coefficients (R) and Spearman *p* values are indicated. **d** Scheme depicting HER2+ 3DiBCs expansion and treatment. Left panel: representative co-culture in a well from 96-well plate. **e** Micrograph of FAP expression in HER2+ 3DiBCs (HCC1954/BF2). Scale bars: 100 μm. Representative of *n* = 3 independent HER2+ 3DiBCs. **f** Bioluminescent tracking of HCC1954 in HER2+ 3DiBCs treated as indicated in absence of ICs. IgG1 was used as control for TTZ. *n* = 9 (HCC1954/BF1; blue) and *n* = 6 (HCC1954/BF2; red) HER2+ 3DiBCs examined per condition, from 3 biologically independent experiments. Values are mean ± s.e.m. **g** Bioluminescent tracking of HCC1954 in HER2+ 3DiBCs treated as indicated in presence of ICs. IgG1 was used as control for TTZ. *n* = 6 (HCC1954/BF1; blue) and n = 9 (HCC1954/BF2; red) HER2+ 3DiBCs examined per condition, from 3 biologically independent experiments. Values are mean ± s.e.m. **h** Representative micrographs of HER2+ 3DiBCs from (**g**). Left panel: TTZ/DP47-IL2v treatment. Right panel: TTZ/FAP-IL2v treatment. Arrows indicate ICs. Scale bar: 20 μm. Red breast cancer cells (BCCs; HCC1954), Green breast fibroblasts (BFs), Cyan immune cells (ICs). Rel relapsing patients, RF relapse-free patients, TTZ trastuzumab, RLU relative luminescence units. Two-sided, unpaired *t* test *p* values (P) are indicated for (**a, f, g**); ns indicates non-significant. Source data are provided as a Source Data file.

immune stromal cells. Indeed, unresponsive HER2+ BC patients' tumors are characterized by an increased expression of CAF-S1, pCAF and F-TBRS signatures. This discovery does not undermine the direct therapeutic cytotoxicity against BCCs nor the development of intrinsic resistance to HER2-targeted therapy. Yet, our findings point to the TME as a major driver of resistance to treatment and argue for the use of stromal cues as early predictors of response to targeted therapy. Therefore, our observations suggesting that trastuzumab-induced ADCC depends on preexisting stromal attributes may improve therapeutic decision-making and current molecular classification of BC. Our conclusions are further supported by various studies that link elevated expression of particular stromal-specific genes with poor outcome in BC[16,42].

Despite the fact that immuno-modulators are being extensively evaluated in clinical trials, the lack of suitable preclinical models is hindering the effective translation of stroma-targeted treatment into the clinical setting. To overcome this obstacle, we have developed an original model that recapitulates ex vivo the immune exclusion that associates with resistance to anti-HER2 mAbs therapy in HER2+ BC.

With about 80% of BFs and 20% of BCCs, our HER2+ 3DiBC model is comparable to stroma-high breast tumors in patients[43]. In this setting, BFs enhanced BCCs proliferation. This may be explain by the fact that fibroblasts from the TME are secreting a plethora of factors that intervene in processes such as cancer cells proliferation, invasion and metastasis[44–46]. Some of these factors include IL-6, IL-8, IL-11, POSTN, FGF2/7, PDGF, HGF, and IGF-2 among others[46–48]. Remarkably, immune cells also stimulated cancer cells growth in an untreated setting. This phenomenon might result from pro-tumorigenic effects of inflammatory cells reported in numerous studies[49,50]. For instance, inflammatory cytokines such as IL-1β, IL-6, IL-17, IL-18, IL-23, and TNF-α can lead to tumor growth through activation of pathways such as NF-κB and STAT3[51–54].

As observed in many cases of breast cancer, our model recreates features of immune evasion that may include T cells and NK cells exclusion. In our experimental setting, this phenomenon is likely due to CAF-associated factors[27,55] since BFs in HER2+ 3DiBC do not tend to self-organize into a physical barrier (Fig. 2a; Fig. 4e, h). T cells were extensively studied in the context of anti-cancer immunity and their presence in the tumor has been associated with better outcome including in HER2+ BC[56,57]. However, CD4+ and CD8+ T cells abundance was not significantly modulated by trastuzumab and FAP-IL2v regimen in our experimental setting. In contrast, NK cells abundance was consistently increased within the immune infiltrate of treated HER2+ 3DiBC, thus suggesting a prominent role of NK cells in driving trastuzumab/FAP-IL2v therapeutic efficiency. In line with these data, NK cells alone were sufficient to recapitulate the anti-cancer effect observed upon trastuzumab/FAP-IL2v treatment. IL2 has drawn increasing attention in oncology[58,59] and several IL2-based compounds[60–62] are currently evaluated in the clinical setting (e.g., NCT02983045,

NCT04009681, NCT04303858, NCT03978689). In this context, our findings derived from modeling HER2+ BC ex vivo shed light on the capacity of IL2-based therapies to enhance trastuzumab effectiveness against BCCs. Altogether, and even though it may only provide an approximation of the complex biological processes taking place in patients' tumors, HER2+ 3DiBC model represents a versatile but also relevant predictive tool allowing rapid and high throughput screening of therapeutic responses.

It is well accepted that CAFs function may be determined by the cell of origin. However, an alternative hypothesis suggests CAF being a cell state depending on autocrine and paracrine signaling[45,63]. Among the major CAFs subtypes recently discovered in BC, CAF-S1 and pCAF subsets were associated with an immunosuppressive microenvironment[27]. We realized that the expression of FAP, a biomarker identifying both CAF-S1 and pCAF subtypes, was modulated by TGF-beta in BFs, thus advocating for CAF-S1/pCAF being a cell state depending on TGF-beta signaling rather than a cell type. Supporting this idea, CAF-S1 and pCAF subsets are also characterized by the increased expression of *SERPINE1*, *IGFBP3* and *PDPN* among others, all of which being upregulated by TGF-beta in CAFs (Supplementary Data 1)[20,27,29]. We observed that BCCs-dependent TGF-beta pathway activation in BFs led to increased FAP expression, thereby corroborating the fact that CAF-S1 and pCAFs are predominantly detected close to epithelial tumor cells in patients[27]. Of note, eight subpopulations of CAFs associated with differential resistance to immunotherapy were identified within the FAP+ CAF-S1 subset[64]. TGF-beta pathway activation in CAFs was also recently associated with immunotherapy failure[65]. In view of our findings, the existence of a heterogeneous population of CAF-S1 may depend on distinct levels of TGF-beta activity, which could have important consequences on treatment election and outcome in BC.

Altogether, our understanding of the clinical response to trastuzumab might strengthen the interest in assessing TGF-beta pathway activation for clinical diagnosis and in using TGF-beta inhibitors in HER2+ BC patients. Indeed, clinical trials are already evaluating the benefit from TGF-beta pathway inhibition in combination with standard regimen in various solid tumors (e.g., NCT04031872, NCT02452008, NCT02937272). However, TGF-beta program activation is highly cell-type specific and impacts epithelial cancer cells as well as every stromal cell permeating the tumor in very different ways. For instance, we discovered that among all stromal TBRSs, F-TBRS was the only genetic program associated with a lack of therapeutic benefit from anti-HER2 mAbs. Therefore, we turned this stromal trait associated to resistance to treatment into a specific therapeutic biomarker. By using the fusion immunocytokine FAP-IL2v, we have proven FAP as an actionable target to direct the otherwise highly toxic immunomodulator IL2 towards unresponsive tumors enriched with

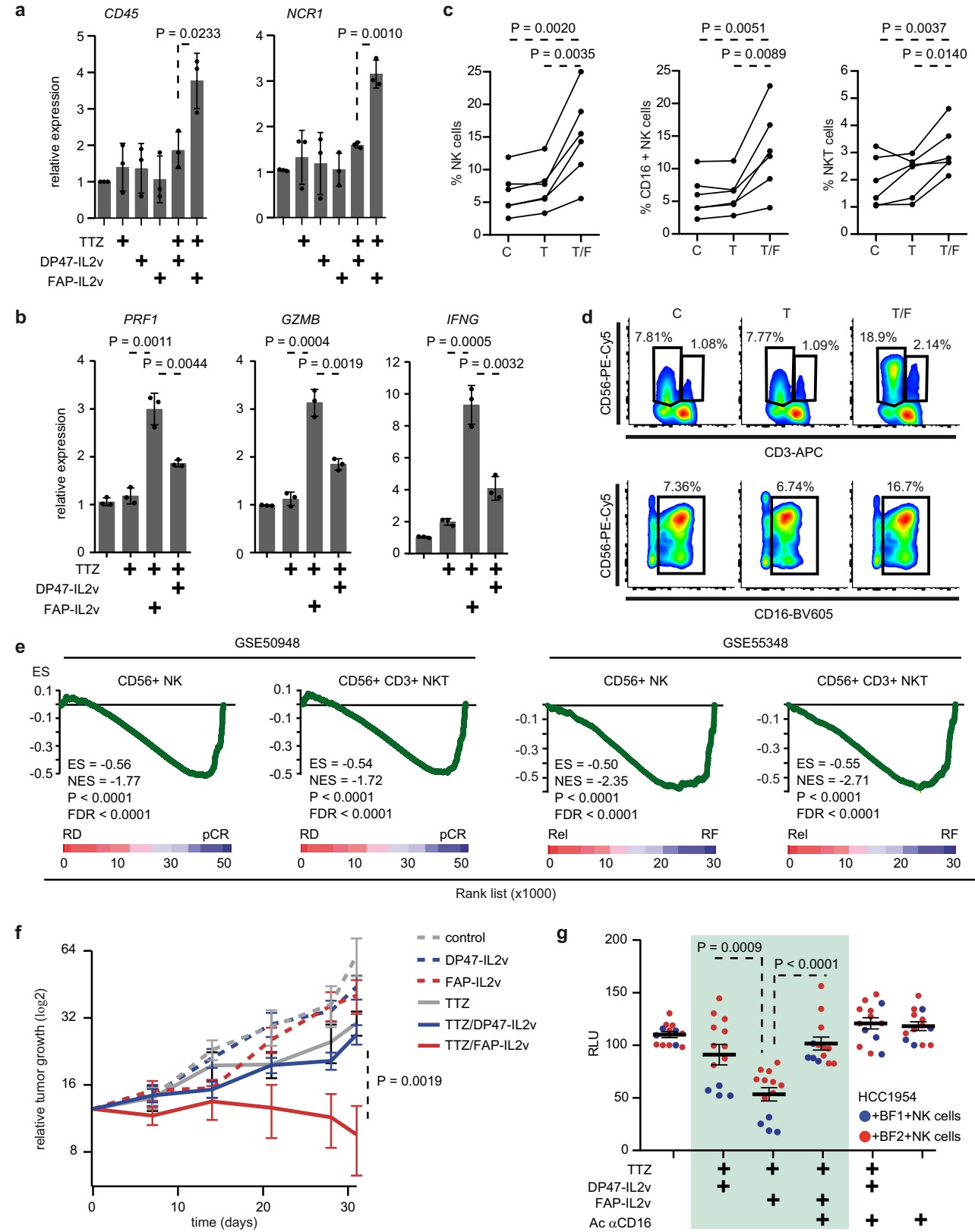

TGF-beta activated stroma. As a result, we reinstated trastuzumab-induced ADCC while avoiding potential adverse-effects due to TGF-beta pathway broad inhibition. Altogether, this work illustrates the potential of targeting non-immune elements in the tumor microenvironment to overcome immune evasion and restore patients' response to treatment.

## Methods

### Study approval
Blood samples of healthy donors and breast cancer patient samples were obtained under informed consent and approval of the MarBio-banc Committee (Parc de Salut Mar; 2017/7294/I) according to Ethical regulations. There was no participant compensation. In vivo

**Fig. 5 | FAP-IL2v enhances ADCC through NK activation. a, b** Relative gene expression levels of (**a**) *CD45* and *NCR1*, (**b**) *PRF1, GZMB and IFNG* measured by qRT-PCR in HER2+ 3DiBCs (HCC1954/BF2) treated as indicated. IgG1 was used as control for TTZ. *n* = 3 biologically independent experiments. Values are mean ± s.d. **c** Flow cytometry analysis using gating panels from Supplementary Fig. 4e and showing percentages of NK cells, CD16+ NK cells and NKT cells in infiltrating ICs from HER2+ 3DiBCs (HCC1954/BF2) treated with trastuzumab (T) or trastuzumab + FAP-IL2v (T/F) compared to control (IgG1; C). *n* = 6 biologically independent experiments. **d** Flow cytometry profiles illustrating data from (**c**). **e** GSEA of CD56+ NK and CD56+CD3+ NKT cells gene signatures comparing pCR vs. RD (left panels) and Rel vs. RF (right panels) HER2+ BC after TTZ-based therapy. **f** Growth kinetics of macroscopic tumors derived from subcutaneous injection of HCC1954 cells into nude mice treated with DP47-IL2v (dashed blue; *n* = 11), FAP-IL2v (dashed red;

*n* = 12), TTZ (gray; *n* = 8), TTZ/DP47-IL2v (blue; *n* = 12), TTZ/FAP-IL2v (red; *n* = 11) compared to control (IgG1; dashed gray; *n* = 7). Values are mean ± s.e.m. **g** Bioluminescent tracking of HCC1954 in HER2+ 3DiBCs treated with TTZ, FAP-IL2v, DP47-IL2v, and/or blocking antibody against CD16 in presence of NK cells. *n* = 4 (HCC1954/BF1; blue) and *n* = 9 (HCC1954/BF2; red) HER2+ 3DiBCs examined per condition, from 3 biologically independent experiments. Values are mean ± s.e.m. pCR pathological complete response, RD residual disease, Rel relapsing patients, RF relapse-free patients, ES enrichment score, NES normalized enrichment score, FDR false discovery rate, TTZ trastuzumab, RLU relative luminescence units. Two-sided, unpaired (**a, b, f, g**) and paired (**c**) *t* test *p* values (P) are indicated. GSEA nominal *p* value (P) and FDR-adjusted *p* value are indicated for (**e**). Source data are provided as a Source Data file.

experiments in murine model were approved by the Animal Research Ethical Committee of Barcelona Biomedical Research Park (CEEA-PRBB; FUE-2018-00801894) and the Catalan government.

## Generation of gene expression signatures and association with clinical parameters

To assess associations between gene expression profiles and clinical information, we used subsets from two publicly available datasets: GSE55348[26] and GSE50948[25]. GSE50948 contains a pool of 48 patients with HER2+ locally advanced breast cancer that received 1 year of treatment with trastuzumab (NOAH trial; 28 RD, 20 pCR). GSE55348 dataset includes disease-free survival information for 51 HER2+ breast cancer patients treated with trastuzumab between 2005 and 2009 for a median period of 1 year (observational study GHEA; 22 Rel, 29 RF). Data were downloaded from GEO microarray data repositories. Pre-processed series matrixes originally provided by the authors were used in the analyses.

TGF-beta response signatures in T cells, macrophages, fibroblasts and endothelial cells, *TGFB*[20] as well as CAF-S1[27], pCAF, and sCAF[29], the CD56+CD3− NK and CD56+CD3+ NKT cells signatures (GSE124395)[37] have been previously described. TGF-beta response in NK cells was derived from NCBI-GEO dataset GSE39197. NK-IL2RS was derived from the GSE12198 NCBI-GEO dataset. NK-IL2RS score discriminating NK-IL2RS high and NK-IL2RS low expression patients was determined using receiver operating characteristic (ROC) curve analysis[66]. Gene lists are provided as a Supplementary Excel File (Supplementary Data 1). In order to enable GSE55348 and GSE50948 analysis, gene sets were transformed into illumina probe sets (Supplementary Data 2) and into affymetrix probe sets (Supplementary Data 3) using g:Profiler (v. 2020-10-12)[67].

ESTIMATE (Estimation of STromal and Immune cells in MAlignant Tumors using Expression data, v. 1.0.13) analysis[24] was performed to infer the presence of infiltrating stromal/immune cells in tumors' transcriptomic data from GSE55348 and GSE50948. A list of enrichment scores related to each immune gene included in ESTIMATE has been added as Supplementary Data 4 for the two analyzed cohorts.

Gene set enrichment analysis (GSEA v. 4.1.0) was performed as previously described to obtain an enrichment score (ES), a normalized enrichment score (NES) which accounts for the size of the gene set being tested, a *p* value and an estimated False Discovery rate (FDR)[68].The ImmuneSigDB was obtained from UC San Diego and Broad Institute[32]. We assessed signatures enrichment scores comparing non-relapsing to relapsing patients after treatment (GSE55348) and patients' response to treatment (GSE50948). We computed *p* values using 10,000 permutations for each signature.

For single gene-set variation analysis (GSVA), GSVA v.1.45.2 package was used as previously described[69]. This function provides data transformation from genes to gene set thereby allowing signature summaries from a gene expression dataset. GSE55348 and GSE50948 datasets were interrogated for generated signatures. Spearman

correlation analyses between signatures were done using ggplot2 v.3.3.6 package[70].

Available annotated clinical data for GSE55348 dataset included disease-free survival intervals. We assessed signature's predictive significance on recurrence with a univariate Cox proportional hazards model likelihood ratio test (SPSS v.22). We obtained Kaplan–Meier survival curves for patients with low and high average signature scores. Statistical significance was assessed by introducing the average signature score as a continuous covariate in the Cox model. Multivariate Cox model included age, gender, histological grade and lymph node status as adjustment covariates. HR (hazard ratio) values correspond to the exponentiation of the odd ratios (risk estimate) for each variable. A value below 1 corresponds to a decreased risk of relapse and a value above 1 indicates an increased risk of relapse. A 95% confidence interval (CI) was displayed for each HR.

## Cells and cell culture

BFs were kindly provided by P. Gascon/P. Bragado laboratory (IDI-BAPS, Barcelona) and are described elsewhere[71]. In brief, BFs were expanded in culture from digested pieces of fresh HER2+ BC tissue. Differential trypsinization was performed to further isolate the fibroblast population. A retroviral vector pMIG (MSCV-IRES-GFP) expressing hTERT and GFP was used to immortalize and label BFs. BFs were characterized by measuring the expression of α-SMA (alpha smooth muscle actin)[71] and FAP (Fig. 4e and Supplementary Fig. 3a). Breast cancer cell lines (SKBR3, ATCC: HTB-30; HCC1419, ATCC: CRL-2326; HCC1954, ATCC: CRL-2338) were provided by the American Type Culture Collection (ATCC, USA). Cell lines used in this study were not listed as known misidentified cell lines by the International Cell Line Authentication Committee. For bioluminescent tracking, cancer cells were engineered to produce a fusion protein reporter construct encoding red fluorescent protein (mCherry) and luciferase (kind gift from E Batlle, IRB Barcelona). In brief, fusion protein reporter construct encoding mCherry and firefly luciferase was cloned into FUW lentiviral vector[72] under control of CMV promoter[20]. Stable expression of luciferase and mCherry was obtained upon lentiviral infection of cancer cells followed by antibiotic selection (Zeocin; InvivoGen, Toulouse, France). Homogenous expression of the reporter was assessed by measuring mCherry fluorescence in infected cells sorted by FACS. TGF-beta activity reporter cells were generated after infection with TGF/SMAD Luciferase Reporter Lentivirus (Kerafast) followed by antibiotic selection (Puromycin; InvivoGen, Toulouse, France). Luciferase reporter assays were performed by adding luciferin potassium salt (Resem BV, Lijnden, Netherlands) at a final concentration of 50ug/ml to cells cultured in 96-well flat bottom clear, white walls plates (Corning Incorporated Life Science, Kennebunk, USA). Bioluminescence was measured following luciferin injection in each well with Microplate Luminometer, Orion II (Berthold Detection Systems, Pforzheim, Germany) and Simplicity 4.2 software. PBMCs (ICs) were purified from blood samples of healthy donors with

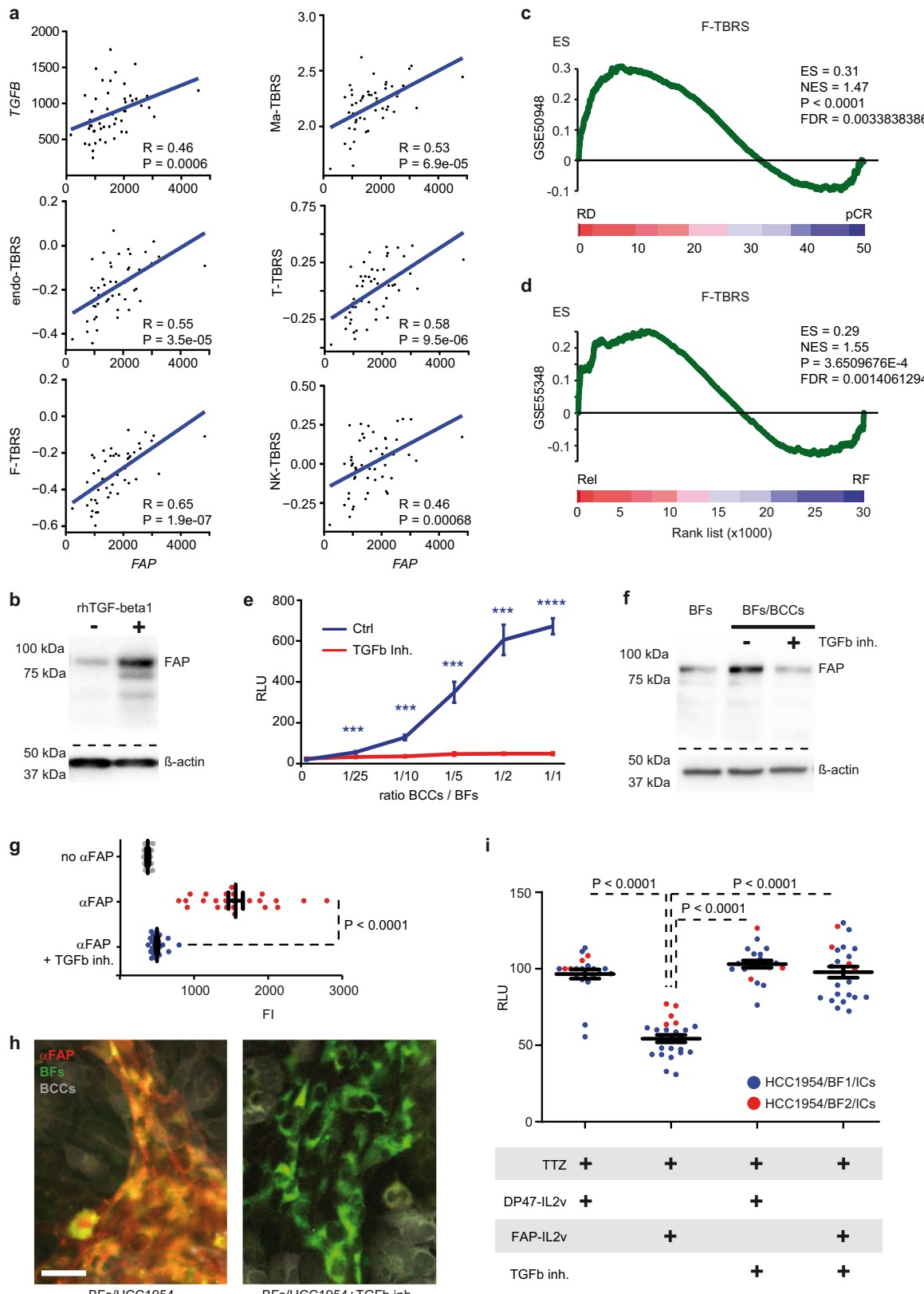

Lymphoprep™ and stained with CellTracker™ Deep Red Dye (Invitrogen) following manufacturers' protocols. Untouched NK cells were purified from PBMCs with the NK Cell Isolation Kit (Miltenyi Biotec) following manufacturer's instructions. NK enrichment was confirmed by cell sorting. Cell lines were tested weekly for mycoplasma contamination and resulted negative.

3D cultures were obtained by mixing fibroblasts and cancer cells in Basement Membrane Extract (BME, Cultrex) and cultured in 0.2% FBS medium. Unless specified otherwise, cells were co-cultured in a 1/5 ratio (BCCs/BFs). Four days after plating, 3D cultures were treated with 150 nM of either anti-FAP-IL2v or untargeted DP47-IL2v antibody (kindly provided by Dr. C. Klein and Dr. V. Teichgräber; Roche

**Fig. 6 | FAP is an actionable biomarker of TGF-beta activity in breast cancer microenvironment. a** Correlation between *FAP*, *TGFB* and TBRSs expression levels in HER2+ BC patients (*n* = 51) from GSE55348 dataset (from up to down, left to right: macrophage, endothelial, T cell, fibroblast, NK cell). Correlation coefficients (R) and Spearman *p* values are indicated. **b** Levels of FAP protein (upper panel) in BFs untreated or treated with recombinant human (rh) TGF-Beta1. Bottom panel shows β-Actin protein levels as normalization control. Representative of *n* = 3 biologically independent experiments. **c, d** GSEA of F-TBRS gene-set comparing (**c**) pCR vs. RD and (**d**) recurring vs. relapse-free HER2+ BC after TTZ treatment. **e** Bioluminescent tracking of TGF-beta pathway activation in BFs co-cultured with increasing amounts of HCC1954, in presence of TGFb inh. (red) compared to untreated control (Ctrl, blue). *n* = 3 biologically independent experiments. Values are mean ± s.d. **f** Level of FAP protein (upper panel) in BFs, BFs co-cultured with BCCs (HCC1954) untreated or treated with TGFb inh. Bottom panel shows β-Actin protein levels as normalization control. Representative of *n* = 3 biologically independent experiments. **g** Mean anti-FAP immunoFluorescence Intensity (FI) detected on BFs co-cultured with HCC1954 in presence (blue) or absence (red) of TGFb inh. compared to controls without anti-FAP antibody (gray). *n* = 25 regions examined per condition, from 5 biologically independent experiments. Values are mean ± s.e.m. **h** Representative micrographs from (**g**). Left panel: anti-FAP antibody identifies BFs. Right panel: loss of anti-FAP binding to BFs upon TGFb inh. treatment. Scale bar: 20 μm. Gray HCC1954 (BCCs), Green BFs, Red anti-FAP antibody immunofluorescence, Yellow FAP/BFs colocalization. **i** Bioluminescent tracking of HCC1954 in HER2+ 3DiBCs with BF1 (blue) or BF2 (red) in presence of ICs and treated as indicated. *n* = 21 (TTZ/DP47-IL2v), *n* = 24 (TTZ/FAP-IL2v), *n* = 21 (TTZ/DP47-IL2v/TGFb inh), *n* = 24 (TTZ/FAP-IL2v/TGFb inh.) HER2+ 3DiBCs examined per condition, from 6 biologically independent experiments. Values are mean ± s.e.m. Rel relapsing patients, RF relapse-free patients, pCR pathological complete response, RD residual disease, ES enrichment score, NES normalized enrichment score, FDR false discovery rate, TGFb inh. TGF-beta pathway inhibitor, RLU relative luminescence units, TTZ Trastuzumab. GSEA nominal *p* value (P) and FDR-adjusted *p* value are indicated for (**c, d**). Two-sided, unpaired *t* test *p* values (P) are indicated for (**e, g, i**); ****P < 0.0001; ***P < 0.001. Source data and exact *p* values for (**e**) are provided as a Source Data file.

Innovation Center, Switzerland) in 0.2% FBS media for 4 h at 37 °C followed by unbound compounds retrieval with three DPBS 0,2% FBS wash steps. Untargeted DP47-IL2v antibody was used to control for the effect of residual unbound compound that main remain after wash steps. Next, ICs or NK cells were added to the 3D culture in 0.2% FBS media w/o mAbs anti-HER2 (trastuzumab). Human IgG1 was used as control antibody. After 72 h co-culture, specific cancer cell expansion was measured by Luciferin (Resem BV, Lijnden, Netherlands)/Luciferase bioluminescence using Microplate Luminometer Orion II (Berthold Detection Systems, Pforzheim, Germany) and Simplicity 4.2 software or by mCherry fluorescence. Fibroblasts and ICs abundance were assessed by GFP and Deep Red fluorescence respectively using an SP5 or SPE confocal microscope (Leica). Microscopy image analysis was performed with Imaris Cell Imaging software (v.9.7, Oxford Instruments) and ImageJ v.1.53i[73].

When indicated, cells were cultured in presence of either TGF-beta1 recombinant protein (Peprotech; 5 ng mL-1) or TGF-beta pathway inhibitor (SB431542, Sigma-Aldrich) at 5 μM concentration. When indicated, anti-CD16 blocking antibody (see Antibodies section) was applied to purified NK cells for 30 min prior to inoculation in HER2+ 3DiBC following manufacturer's instruction.

**Clinical material**
Biological samples were obtained from 22 HER2+ BC patients treated with anti-HER2 therapy (13 patients responsive and 9 unresponsive to treatment; see Supplementary Table 1). Samples were collected within the usual clinical practice. Clinical information was anonymized by medical doctors collaborating to the project. International standards of Ethical Principles for Medical Research Involving Human subjects (code of ethics, Declaration of Helsinki, Fortaleza, Brazil, October 2013) were followed in accordance with legal regulations on data confidentiality (Organic Law 3/2018—December the 5th—on the Protection of Personal Data and Digital Rights Guarantee) and on biomedical research (Law 14/2007—July the 3rd).

**Immunohistochemistry**
Immunostainings were carried out using 4 μm tissue sections according to standard procedures. Briefly, after antigen retrieval, samples were blocked with Peroxidase-Blocking Solution (Dako, S202386) for 10 min at RT. Primary antibodies (see Supplementary Table 2) were incubated o/n at 4oC. Slides were washed with EnVision™ FLEX Wash Buffer (Dako, K800721). Corresponding secondary antibody was incubated with the sample for 45 min at RT. Samples were developed using 3,3'-diaminobenzidine, counterstained with hematoxylin and mounted. Staining analyses were performed with QuPath software v.0.3.2[74] and by histological scoring (H-Score)[75].

**Cell sorting**
In order to investigate the immune infiltrate in HER2+ 3DiBC, non-infiltrating immune cells were excluded prior to analyses as follow. Three DPBS wash were performed after culture media retrieval. BME drops containing the immune infiltrate were then pulled out from the culture well with a cut-off P1000 pipet tip and collected in 500 μl trypsin. Resulting single cells preparation was pooled into cold 10% FBS medium and processed for immunostaining with anti-CD45, CD3, CD4, CD8, CD20, CD14, CD16, CD56 antibodies mix (see Antibodies section) for 30 min on ice in the dark following manufacturer's protocol. In all flow cytometry experiments, initial gating was performed as follow: (1) FSC-A/SSC-A: debris exclusion; (2) FSC-A/DAPI: viable cells selection; 3) FSC-A/FSC-H: single cells selection. Further gating strategy is provided in Supplementary Fig. 4b, e. For FAP analyses, single cells suspension of either BFs, BCCs or ICs were incubated with anti-FAP 4B9 antibody for 30 min on ice. Detection was achieved with Alexa 790-coupled secondary anti-human antibody (see Antibodies section) following manufacturer's protocol. Cells were washed in 2 ml of cold 2% FBS DPBS and centrifuged at 400 g/ 10 min/4 °C. Cell pellets were resuspended in 2% FBS DPBS. Cell suspension was stained with DAPI. Cell sorting and analysis was performed in a FACSAria II SORP cytometer (BD Biosciences). Compensation was performed using negative control samples and single positive controls. Post-sort analysis was performed to determine the purity of sorted cells prior to RNA extraction. Data analysis was performed with Flow Jo v.10.8 software. Abundance of relevant cell populations within post-sort fractions is provided in Supplementary Data 5.

**Quantitative RT-PCR**
RNA was extracted using Trizol Reagent (Invitrogen) and isolated with the RNeasy Micro Kit (QIAGEN, Hilfen, Germany) following manufacturer's handbook. Reverse transcription was performed using High Capacity cDNA Reverse Transcription Kit (Applied Biosystems, Thermo Fisher Scientific, Pleasanton, CA). Quantitative PCR was performed using TaqMan assays (Applied Biosystems; *FAP*: Hs00990806_m1, *CD45*: Hs00236304_m1, *NCR1*: Hs00183118_m1, *PRF1*: Hs00169473_m1, *GZMB*: Hs01554355_m1, *IFNG*: Hs00989291_m1, *ERBB2*: Hs01001580_m1, *IL2RB*: Hs01081697_m1, *TGFB1*: Hs00998133_m1, *PPIA*: Hs99999904_m1) following manufacturer's instructions in a 7900HT Fast Real-Time System (Applied Biosystems). For immune infiltrate examination in HER2+ 3DiBC, non-infiltrating immune cells were excluded prior to analyses as mentioned above (see Cell sorting section).

**Orthotopic mouse studies**
Animals were maintained in specific pathogen-free conditions with controlled temperature/humidity (22 °C/55%) environment on a 12-

h light-dark cycle and with food and water *ad libitum*. $0.3 \times 10^6$ HCC1954 cells were inoculated subcutaneously in 5–6-week-old swiss nude female mice (Strain #: 002019; Jackson Laboratories). Mice bearing macroscopic tumors (average tumor size 50 mm³) were randomly assigned to experimental groups and injected once per week with either control IgG1 (4 mice; 7 initiated tumors prior to treatment), trastuzumab (4 mice; 8 initiated tumors prior to treatment), FAP-IL2v (6 mice; 12 initiated tumors prior to treatment), DP47-IL2v (6 mice; 11 initiated tumors prior to treatment), trastuzumab/DP47-IL2v (6 mice; 12 initiated tumors prior to treatment) or trastuzumab/FAP-IL2v

(6 mice; 11 initiated tumors prior to treatment) as previously described[76]. Animals were caged together and treated in the same way. Nor the technician or the investigator could distinguish them during the experiment or when assessing outcomes. All treatments were administered by intravenous injection. Tumor expansion was assessed by caliper measurements. Animals' general condition was monitored using fitness and weight control throughout the experiment. Due to deteriorating clinical alterations (abnormal posture, lack of mobility, weight loss) or absence of tumor development, two animals were excluded from the study. In both cases, exclusion occurred prior to treatment initiation and did not lead to data exclusion. Maximum tumor size (1500mm³) permitted by ethics committee was not exceeded. At experimental end point, mice were euthanized in a chamber with saturated CO2 atmosphere. Euthanasia was confirmed by cervical dislocation.

## Immunofluorescence

Anti-FAP antibody clone 4B9 was kindly provided by Dr. C. Klein and Dr. V. Teichgräber (Roche Innovation Center, Switzerland). Anti-FAP antibody (see Supplementary Table 2) was incubated o/n at 4 °C. Secondary antibody was incubated for 45 min at room temperature in the dark. Cell nuclei were stained with DAPI and slides were mounted in Glycerol/PBS/Phenylenediamine for observation using an SP5 or SPE confocal microscope (Leica).

## Western blot

Protein extracts were obtained by lysing cells in 1 mM EDTA, 1 mM EGTA, 1% SDS. Protein concentration was measured using the Bio-rad kit Protein Assay. Proteins were separated by SDS gel electrophoresis and transferred to PVDF membrane (Millipore). Primary antibodies (see Supplementary Table 2) were incubated o/n at 4 °C in 5% BSA in TBS-Tween 0.1% (blocking solution). Secondary antibodies (1/10,000 dilution) coupled to peroxidase were incubated for 1 h in blocking solution. Membranes were washed in TBS-Tween 0.1%. Immunocomplexes were detected using ECL kit (Amersham International).

## Statistics

Sample size was chosen following previous experience in the assessment of experimental variability (generally all measurements were performed with $n \geq 3$ biological replicates). Statistical analyses of between-group differences were performed using Student's *t* test (Graphpad Prism 8.0.1). Two-tailed *p* values < 0.05 were considered significant.

## Reporting summary

Further information on research design is available in the Nature Research Reporting Summary linked to this article.

## Data availability

GSE55348, GSE50948, GSE39197, and GSE12198 datasets used in this study are publicly available in the NCBI-GEO database. The data are available within the Article, Supplementary Information, Supplementary Data and the Source Data file provided with this paper.

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

## Acknowledgements

This work has been supported by grants from Instituto de Salud Carlos III (ISCIII) co-funded by the European Union (PI17/00211, PI20/00011, CP16/00151; Spanish Ministry of Economy and Competitiveness) to A.C. This work was also supported by ISCIII (CIBERONC CB16/12/00241, PI18/00006, PI21/00002), Generalitat de Catalunya (no. 2017 SGR 507) and the European Community through the Regional Development Funding Program to J.A. A.C. is the recipient of Miguel Servet research contracts from Instituto de Salud Carlos III co-funded by the European Union (MS16/00151, CPII21/00012). E.R. fellowship was funded by the Fundació Privada Cellex. J.L. is the recipient of a Junior Clinician fellowship from the Spanish Association against Cancer (AECC; 6884).

## Author contributions

A.C. and J.A. conceived and designed the study. E.R. and J.L. set up the experimental models and performed treatment analyses. C.M.C., O.B., and N.T. gathered clinical data and performed histopathological analyses. A.G.L. and J.L. performed immune cells purification. J.L., E.R., M.Z., and A.S.A. performed co-cultures and RNA/protein analyses. A.La, L.B., and A.Ll performed fluorescence microscopy and bioimaging. E.R. and J.B.R. performed flow cytometry analyses. I.P.N. and T.C.T. performed GSVA and correlation analyses. M.Ga and A.S.A. performed immunostainings experiments. R.G. and M.Gu designed and performed in vivo experiments. E.R., J.L., J.A., and A.C. interpreted and discussed the results. A.C. performed transcriptomic analyses and wrote the paper. All authors approved this paper for publication.

## Competing interests

J.A. has received consulting fees and honoraria from Seagen, Pfizer, AstraZeneca, Lilly, Merck, Roche, Gilead, Novartis and Daiichi-Sankyo, receives royalties from a licensed patent to Biocartis (EP11382270.4; Mutations in the epidermal growth factor receptor gene) and holds stock options from Inbiomotion. Other authors declare no competing interests.

## Additional information

[1]Cancer Research Program, Hospital del Mar Medical Research Institute (IMIM), Barcelona, Spain. [2]Institute for Research in Biomedicine (IRB Barcelona), Barcelona Institute of Science and Technology (BIST), Barcelona, Spain. [3]Institute for Bioengineering of Catalonia (IBEC), Barcelona Institute for Science and Technology (BIST), Barcelona, Spain. [4]Department of Medical Oncology, INCLIVA Biomedical Research Institute, University of Valencia, Valencia, Spain. [5]Medical Oncology Department, Hospital del Mar, Barcelona, Spain. [6]Universitat Pompeu Fabra, Barcelona, Spain. [7]Centro de Investigación Biomédica en Red de Oncología (CIBERONC-ISCIII), Madrid, Spain. [8]These authors contributed equally: Elisa I. Rivas, Jenniffer Linares. [9]These authors jointly supervised this work: Joan Albanell, Alexandre Calon. ✉e-mail: acalon@imim.es

