## [Peer Review File · Nature Communications]

Reviewers' Comments:

Reviewer #1:

Remarks to the Author:

This manuscript describes findings of HER2+ breast tumors, obtained by analyses of patient materials (IHC) and patient data (mainly GSEA), and by a 3D spheroid system that contained HER2+ human breast cancer (BC) cell lines + BC patient-derived fibroblasts + PBMC from healthy donors, analyzed in the presence or absence of trastuzumab (TTZ) and IL-2. When all data are taken together, the overall message that the authors are trying to make is that fibroblasts inhibit the ability of immune cells – presumably NKs – to exert trastuzumab (TTZ)-mediated ADCC against HER2+ BC cells, where ADCC responses are increased by IL-2; they also addressed the roles of TGFbeta (TGF) in regulating some of these processes.

Although the manuscript presents many data, the paper is very fragmented; it actually includes three parts that are not well-connected to each other. The fibroblast theme was supposed to connect between the different parts of the study, but in several experiments the roles of fibroblasts in the system are not well controlled. Also, the authors reach the conclusion that NK cells are involved in tumor cell killing and that they act by ADCC. This is a repeated statement in the paper, despite the fact that NK cells could act by granzyme/perforin- or FAS/FASL-induced mechanisms in this allogeneic system. They also did not address the possible roles of T cells in regulating the processes addressed in the 3D spheroid experiments.

Major comments:

1. Conclusions made by the authors, based on findings obtained with the 3D spheroid system:

Many of the findings were obtained with this system, using human PBMC – termed herein ICs (immune cells) – from healthy individuals that were added to human HER2+ BC cell lines and human fibroblasts. In all experiments except for Figure 5g, the ICs were not separated to specific subsets of mononuclear cells; rather, they could contain different types of T cells, B cells, monocytes, NK cells and NKT cells.

In view of the complex nature of the IC preparations that were used, it is unfortunate that the authors have ignored the potential involvement of T cells in their system. Also, they have not considered non-ADCC activities of NK cells in this system, that could have resulted from its allogeneic nature.

Here, several examples are given to alternative/additional scenarios that may be involved in the regulation of the responses the authors described:

A. T cells:

As expected, the IC preparations used in the study contained CD4+ and CD8+, as demonstrated in Supplementary Figure 4d. Being an allogeneic system, these T cells are expected to attack the tumor cells because they recognize them as “foreign”.

In contrast to this expected scenario, Figure 2C demonstrates that ICs have increased the growth of BC cells. These findings could be the result of the activities of gamma-delta T cells and the inflammatory factors they release. Such cells were reported as being able to elevate the growth of tumor cells, including BC cells (Front. Imm. 11:2186, 2020; J. Imm. 187:1031, 2011). Moreover, the 2011 report demonstrated that under certain conditions – which are very relevant to the current study because they included TTZ in HER2+ BC – gamma-delta T cells could act differently and promote the anti-tumor activities of (J. Imm. 187:1031, 2011).

Currently, the authors do not mention at all the possibility that T cells may be involved in the effects they measured, and put NK cells at the center of stage. Even if experiments with T cells are not performed, the authors should discuss their potential roles in the process.

B. NK cells

Throughout the paper, the authors claim very strongly that NK cells acted via ADCC to kill the tumor cells in the 3D spheroid system. This statement repeated several times in the manuscript, ignoring the possibility that in this allogeneic system other mechanisms could stand in the basis of NK cell activities.

In immune activities, the attack of normal cells by NK cells is inhibited when normal cells express autologous MHC molecules. In contrast, in the allogeneic setting of the present study, NK cells may attack the tumor cells because they express allogeneic and not autologous MHC molecules or because they do not express MHC at all (which is often the case in tumor cells).

The implication in such a case is that NK cells may kill the tumor cells by non-ADCC-mediated mechanisms, but rather by granzyme/perforin- or FAS/FASL-mediated mechanisms. Such non-ADCC NK cell activities could explain the decrease noted in Figure 4e in cancer cell growth (when fibroblasts were present) in the presence of FAP-IL2v + ICs, without TTZ, compared to the two left control groups. The addition of TTZ further added to the killing effect, but one cannot rule out the possibility that the impact of FAP-IL-2v + ICs + TTZ actually reflected the joint activities of ADCC + non-ADCC NK activities.

2. The fragmented and non-consistent nature of the study:

The study is fragmented in two ways:

A. Fibroblasts are those that are supposed to connect between the different parts of the study, but this is actually not the case.

In general, the study has three parts.

- Part 1: Studies of the impact of fibroblasts on tumor cell killing, in the setting of TTZ+ICs.
- Part 2: Studies of IL-2 in advancing the killing effect; the authors have used a FAP-IL2v construct, in which IL-2 was conjugated to antibodies to FAP, a molecule that is typically expressed by CAF-S1 cells.
- Part 3: Studies of TGF, suggesting that TGF induces FAP expression by fibroblasts, that then binds FAP-IL-2v.

Despite the fact that fibroblasts are the linking theme of the study, the experiments of Part 2 did not always control for the effect of fibroblasts. To give an example: The experiments of Figures 4e demonstrated that FAP-IL2v elevated the efficacy of IC-mediated killing of BC cells in the presence of TTZ, in a fibroblast-containing setting. However, it is not clear if FAP-IL2v indeed bound to fibroblasts, leading then to IL-2-induced effects. The control of similar experiments without fibroblasts should have been performed. The same is correct for the findings described in Figure 5g.

The opposite experimental approach was taken by the authors in the in vivo experiments of Figure 5f, where the experiments were performed ONLY in the absence of fibroblasts, thus the important aspect of response in the presence of fibroblasts was missing.

Concerning Part 3: Many aspects of the TGF story were not completed and the connection of this part to Parts 1 and 2 is loose.

B. The use of 3 HER2+ BC cell lines, in a non-consistent manner:

The 3D spheroid system has used 3 different human BC cell lines. However, in different sections of the Figures, different cell types were used. Moreover, in Figs. 2a and 2e, Figs. 3e and 3f and Figs 4d and 4e, the type of cell line used was not indicated.

Figure 2c demonstrates results of SKBR3 and HCC1954 cells, obtained in the 3D spheroid system. The next aspect of the 3D spheroid system – in Figures 2e, 2f, 3e and 3f – was performed with one of the cell lines, and it is not indicated which one it was. In the follow up part, in Figures 4d and 4e, it is also not clear which two cell lines were used. The next stage of the study was performed with SKBR3 cells (Figs 5a-5d). Then, the study was continued to the TGF aspects, performed with HCC1954 cells.

Although parts of some of the aspects were analyzed in additional cell lines and were presented in the Supplementary materials, the overall flow of study is inconsistent and hard to follow. It is also very difficult to grasp how the observations of one section relate to other sections.

3. Findings on TGF:

In Part 3 of the study, the authors suggest that TGF induces FAP expression by fibroblasts, that then binds FAP-IL-2v.

If indeed so, it would have been expected that tumor cell killing in 3D spheroids of BC cells + ICs + fibroblasts + FAP-IL2v will be stronger than in 3D spheroids not containing fibroblasts, but this comparison was not performed.

Moreover, the authors have not shown directly that FAP is needed for tumor cell killing in this setting, e.g. by knocking-down/out FAP expression in the fibroblasts. The evidence they have provided was indirect through the use TGF inhibition that has led to reduced expression of FAP.

Additional comments:

1. Title:

The authors have demonstrated mechanisms that control IC-mediated killing of HER2+ BC cells, but they did not reach the point of “predicting” resistance of HER2+ patients to TTZ treatments. This may be a possible implication of the study, but the title should describe the findings and not their possible implications.

2. IHC patient analysis of Figure 1:

CD45 expression signifies many immune cell types that may have opposing effects on malignancy. To connect patient IHC data to the study, analysis of NK cells should have been performed and presented in Part 2 of the study.

3. The immune gene signature of Figure 1:

The signature contains genes with opposing roles in regulation of cancer progression. Did the authors try to dissect the GSEA analysis into sub-groups of genes, in effort to achieve some kind of dissociation between genes that have been typically classified as malignancy-supporting genes from genes that are typically identified with anti-malignancy functions?

4. Figure 2c, Figures 4d and 4e:

The data of two different cell lines were combined in one statistical analysis. This is a non-conventional mode of analysis. Each cell line should be analyzed alone (HCC1954 separately from SKBR3).

5. Figure 3c:

What is the number of patients in each group of the cohort?

How did the authors determine NK-IL2RS high vs. NK-IL2Rs low?

6. Figure 4a:

The Figure demonstrates FAP expression in HER2+ BC patients vs normal breast. It is not tightly connected to the study because it does not present data connected to TTZ treatment.

How many patient samples were analyzed by IHC for FAP expression?

7. Figure 5a:

Control of DP47-IL2v is missing.

8. Figure 5g (and 5f):

The figure was supposed to demonstrate the activities of NK cells, but the text inside the figure states "ICs" and not "NK". The legend to Figure 5g does not clarify this inconsistency, because the description of part g is missing. The legend of part f is also missing.

9. The fibroblast lines BF1 and BF2:

Were they CAFs or NAFs? How were they validated? How were they immortalized?

10. The term "infiltration":

The term "infiltration" was often used by the authors to describe the increased presence of immune cells in the 3D spheroid system, in the presence of IL-2. However, IL-2 could have led to elevated proliferation of activated T cells and of NK cells. Thus, the term should be changed to "infiltration/proliferation".

Reviewer #2:

Remarks to the Author:

In this study, the authors study report the development of a fully humanized immunocompetent ex vivo method of HER2+ breast cancer cells, cancer-associated fibroblasts (CAFs) and immune cells that recapitulates patients' response to trastuzumab. The authors further identify distinct stromal traits predicting the lack of benefit from trastuzumab therapy. Notably, stroma-targeted stimulation of IL-2

pathway with Simlukafusp Alfa, an immunocytokine composed of IL2v fused to a FAP-targeting antibody (FAP-IL2v) currently in clinical trial, restored trastuzumab activity in unresponsive tumors.

This study is well-conducted, well-controlled and data for the most part support the conclusions. The study is also very timely and will certainly stimulate broad interest in the field on immuno-oncology, especially considering the renewed interest in IL-2: currently at least 11 candidate drugs based on engineered IL-2 are currently at IND stage or in clinical trials (Nat Rev Drug Discovery 20, 163-165; 2021).

Below are my specific comments:

1. The abstract should mention that the immunocompetent model used is an ex vivo model.

2. Figure 1: Did you use the ESTIMATE method from ref 43?

3. Figure 2b: Which factor(s) expressed by BFs is (are) responsible for increased BCCs proliferation. Please show correlation (in supplementary) between luciferase expression and cell proliferation/cell death, as this is the main read-out.

4. Please add methodological details regarding the plasmid used to engineer bioluminescent/fluorescent cancer cells and how the luciferase assay was performed.

5. Figure 2d: Please discuss how immune cells (IC) from naive patient may increase BCCs' proliferation.

6. Figure 2c-f: Please indicate the ratio of BCC/BFs used in the figure legends. Is the ratio of BCC/BFs used in all experiments representative of the physiological ratio found in primary tumors and/or metastatic lesions?

7. Can you assess the impact of BCC/BFs ratio on TTZ efficacy and ICs infiltration? Are ICs excluded from the spheroids because the number of fibroblasts is too important (i.e. acts as a physical barrier)?

8. Figure 3 conclusion (line 170-173): please clarify how data implies 'the existence of specific stromal signaling associated with resistance to anti-HER2 mAbs'. This could be made more clear in the text.

9. Line 184: "thus confirming our observations in patients". It is not clear which observations in patients you are referring to.

10. Figure 5: Please explain how you discriminated between the immune infiltrate inside of the

spheroids vs immune cells excluded from the spheroids, for qPCR or FACS analysis? If you are not measuring immune cells infiltration per se, then the text should be modified accordingly.

11. Figure 5a-b: Can you comment on the very interesting difference between FAP-IL2v and DP47-IL2v? Cytotoxic activity is improved when the IL2 is targeting the fibroblasts, even within this in vitro setting. In figure 4a, FAP-IL2v increased CD45+ cells "infiltration" (or expansion), can you assess whether DP47-IL2v equally increased CD45+ mRNA level within the spheroids or this is dependent on the fibroblast targeting of IL2?

12. Figure 5c-d: The frequency of other immune cell types (CD4, CD8, monocytes, B cells) should be represented (in supplementary).

13. Figure 5f-g: Legend is missing/incorrect. Panel g: were purified NK cells used? If so, it should be indicated in the figure legends.

14. Line 533-534 (Methods). What clinical alterations did you observe that led to mouse exclusion from the study? Did mouse exclusion lead to data exclusion? The total number of mice per group and number of mice excluded should be specified.

15. Figure 6: Where is the TGF- β coming from in your ex vivo model and in HER2+ cancer. For the moment, the production of TGF- β by BCCs is only presumed.

16. Can you comment on the possible effect of FAP-IL2v on CD8+T cell activity?

17. In the discussion, it would be appropriate to briefly mention the other engineered IL-2 variants currently in clinical trials (Bempegaldesleukin, Nemvaleukin alfa, SAR444245, RG6279, CUE-101).

18. Consider changing the title to better reflect findings. 'Tumor microenvironment examination...' is quite vague.

Reviewer #3:

Remarks to the Author:

About 50% of HER2+ breast cancer (BC) patients do not benefit from HER2-targeted therapy and almost 20% of them would relapse after treatment. This study identified distinct stromal traits predicting the lack of benefit from therapy. The authors observed that TGF- β -activated CAF, specifically, CAF-S1 subpopulation were increased in tumors resistant to therapy, and associated with low activity of IL2. Stroma-targeted stimulation of IL2 pathway in unresponsive tumors restored trastuzumab anti-cancer efficiency, and FAP-IL2v may restore trastuzumab-mediated ADCC against BCCs in unresponsive HER2+ BC. The topic of the manuscript is interesting and has clinical significance.

Effect of IL2 in combination of trastuzumab in MBC was contractively reported by different studies. Results from phase II trial of trastuzumab in combination with low-dose IL2 in patients with metastatic breast cancer who have previously failed trastuzumab did not find clinical benefit and lack of expansion of NK cells in response the treatment (Mani et al. Breast Cancer Res Treat. 2009 Sep; 117(1): 83-89). Currently, FAP-IL2v is in Phase 1b/2 clinical development as a monotherapy and in combination with trastuzumab and cetuximab. The finding from this study could advance the understanding how FAP-IL2v functional enhance the therapeutic effect of Transtuzumab, and which would have better benefit to the treatment.

Overall, the manuscript is prepared well, and the approaches used in the study are appropriate. The mouse model they developed for the study is novel. The data presented is clear with good quality figures.

Reviewer #4:

Remarks to the Author:

The authors of this manuscript propose that resistance to trastuzumab in HER2 breast cancer may be mediated by stromal cells, in particular CAF-S1 cells activated by TGF β and that this resistance can be reversed by TGF β inhibitors. They also attempt to show that NK cells are important in response to trastuzumab that may be mediated by IL-2 activity. The following comments and suggestions are provided:

1. The authors show that naive ICs are somehow stimulatory to tumor growth. Is there evidence for this in the literature?

2. In figure 1d the authors show that a stromal gene signature is associated with resistance to trastuzumab in 2 clinical datasets. Why is it then necessary to invoke the CAF-S1 subset when a generalized stromal signature and unsorted fibroblasts that are mixed with BC are shown to produce this effect? Were other CAF subtypes examined?

3. The authors indicate that NKIL2RS explains about 70% of cancer

169 relapses after HER2-targeted therapy independent of the main clinical variables, however, this is not consistent with the data nor can this sort of extrapolation be made. Also in the same table, it is unclear why the difference between low (.784) and high (6.336) grade is not significant whereas the difference between low .096 and high (.606) NKIL2RS is highly significant.

4. On page 6, paragraph 2 it is indicated that the FAP protein is associated with expression of CAF-S1 and inversely associated with NKIL2RS yet FAP is not associated with response in the two clinical cohorts. This seems counterintuitive is there an explanation for this?

5. There is no explanation for Fig 5f+g in the legend. Please clarify.

6. Under methods please describe how fibroblasts were isolated and characterized.

Revised manuscript submission, NCOMMS-21-28401A

Response to reviewers' comments

We thank the reviewers for their insightful comments that have improved this revised version to a large extent. You'll find below a point-by-point reply detailing how we have addressed each of the reviewers concerns. We have labeled new data in blue in the text.

Reviewer #1 (Remarks to the Author):

This manuscript describes findings of HER2+ breast tumors, obtained by analyses of patient materials (IHC) and patient data (mainly GSEA), and by a 3D spheroid system that contained HER2+ human breast cancer (BC) cell lines + BC patient-derived fibroblasts + PBMC from healthy donors, analyzed in the presence or absence of trastuzumab (TTZ) and IL-2. When all data are taken together, the overall message that the authors are trying to make is that fibroblasts inhibit the ability of immune cells – presumably NKs – to exert trastuzumab (TTZ)-mediated ADCC against HER2+ BC cells, where ADCC responses are increased by IL-2; they also addressed the roles of TGFbeta (TGF) in regulating some of these processes.

Although the manuscript presents many data, the paper is very fragmented; it actually includes three parts that are not well-connected to each other. The fibroblast theme was supposed to connect between the different parts of the study, but in several experiments the roles of fibroblasts in the system are not well controlled. Also, the authors reach the conclusion that NK cells are involved in tumor cell killing and that they act by ADCC. This is a repeated statement in the paper, despite the fact that NK cells could act by granzyme/perforin- or FAS/FASL-induced mechanisms in this allogeneic system. They also did not address the possible roles of T cells in regulating the processes addressed in the 3D spheroid experiments.

Major comments:

1. Conclusions made by the authors, based on findings obtained with the 3D spheroid system:

Many of the findings were obtained with this system, using human PBMC – termed herein ICs (immune cells) – from healthy individuals that were added to human HER2+ BC cell lines and human fibroblasts. In all experiments except for Figure 5g, the ICs were not separated to specific subsets of mononuclear cells; rather, they could contain different types of T cells, B cells, monocytes, NK cells and NKT cells.

In view of the complex nature of the IC preparations that were used, it is unfortunate that the authors have ignored the potential involvement of T cells in their system. Also, they have not considered non-ADCC activities of NK cells in this system, that could have resulted from its allogeneic nature.

Here, several examples are given to alternative/additional scenarios that may be involved in the regulation of the responses the authors described:

A. T cells:

As expected, the IC preparations used in the study contained CD4+ and CD8+, as demonstrated in Supplementary Figure 4d. Being an allogeneic system, these T cells are expected to attack the tumor cells because they recognize them as “foreign”.

In contrast to this expected scenario, Figure 2C demonstrates that ICs have increased the growth of BC cells. These findings could be the result of the activities of gamma-delta T cells and the inflammatory factors they release. Such cells were reported as being able to elevate the growth of tumor cells, including BC cells (Front. Imm. 11:2186, 2020; J. Imm. 187:1031, 2011). Moreover, the 2011 report demonstrated that under certain conditions – which are very relevant to the current study because they included TTZ in HER2+ BC – gamma-delta T cells could act differently and promote the anti-tumor activities of (J. Imm. 187:1031, 2011).

Currently, the authors do not mention at all the possibility that T cells may be involved in the effects they measured, and put NK cells at the center of stage. Even if experiments with T cells are not performed, the authors should discuss their potential roles in the process.

In agreement with this reviewer’s comments and observations derived from HER2+3DiBC model, immune cells can contribute to tumor growth, probably through the establishment of a beneficial inflammatory niche. In this line, pro-tumorigenic inflammation has become one of the hallmarks of cancer (Hanahan & Weinberg, 2011), as numerous studies have shown that tumor development depends on interactions between cancer cells and inflammatory cells through secretion of cytokines. For instance, IL-1 β and IL-18 secreted in response to cellular stress has been shown to trigger tumor progression through activation of NF- κ B (Van Gorp & Lamkanfi, 2019). Other cytokines such as TNF- α , IL-6, IL-17 and IL-23 may lead to tumor growth through activation of inflammatory pathways such as NF- κ B and STAT3 (Liubomirski *et al*, 2019; Zamarron & Chen, 2011; Chabab *et al*, 2020).

In addition and as observed in many cases of breast cancer, our model recreates features of immune evasion that may include T cells exclusion which is likely due to CAF-associated factors (Salmon *et al*, 2019; Costa *et al*, 2018). T cells were extensively studied in the context of anti-cancer immunity and have been shown to improve disease outcome including in HER2+ BC (Capietto *et al*, 2011; Stanton & Disis, 2016). However, CD4+ and CD8+ T cells expansion/infiltration was not significantly modulated by trastuzumab and FAP-IL2v regimen in our experimental setting (supplementary figure 4f). In contrast, NK cells were constantly increased within the immune infiltrate of treated HER2+3DiBC (figure 5c, supplementary figure 5a), thus suggesting a prominent role of NK cells in driving trastuzumab/FAP-IL2v therapeutic efficiency. In line with these data, NK cells alone were sufficient to recapitulate the anti-cancer effect observed upon trastuzumab/FAP-IL2v treatment (figure 5g).

We have now included data and relevant references in the manuscript to address this reviewer's concerns.

B. NK cells

Throughout the paper, the authors claim very strongly that NK cells acted via ADCC to kill the tumor cells in the 3D spheroid system. This statement repeated several times in the manuscript, ignoring the possibility that in this allogeneic system other mechanisms could stand in the basis of NK cell activities.

In immune activities, the attack of normal cells by NK cells is inhibited when normal cells express autologous MHC molecules. In contrast, in the allogeneic setting of the present study, NK cells may attack the tumor cells because they express allogeneic and not autologous MHC molecules or because they do not express MHC at all (which is often the case in tumor cells).

The implication in such a case is that NK cells may kill the tumor cells by non-ADCC-mediated mechanisms, but rather by granzyme/perforin- or FAS/FASL-mediated mechanisms. Such non-ADCC NK cell activities could explain the decrease noted in Figure 4e in cancer cell growth (when fibroblasts were present) in the presence of FAP-IL2v + ICs, without TTZ, compared to the two left control groups. The addition of TTZ further added to the killing effect, but one cannot rule out the possibility that the impact of FAP-IL-2v + ICs + TTZ actually reflected the joint activities of ADCC + non-ADCC NK activities.

We fully agree with this reviewer, HER2+3DiBC is a humanized immunocompetent *ex vivo* model that may only provide an approximation of the complex biological processes taking place in patients' tumors. As pointed out, the decreased BCCs growth observed upon

untargeted DP47-IL2v or FAP-IL2v monotherapy suggest some levels of IL2v activity *per se* in our experimental setting. However, superior anti-cancer activity was observed upon trastuzumab/FAP-IL2v regimen compared to control trastuzumab/DP47-IL2v in presence of ICs (figure 4g, 6i, supplementary figure 3e) or NK cells (figure 5g, supplementary figure 5e). This anti-cancer activity was associated with increased expression of granzyme B and perforin 1 (figure 5b, supplementary 4a) that are among the downstream effectors of ADCC (Capuano *et al*, 2021; Nigro *et al*, 2019; Sordo-Bahamonde *et al*, 2020). We have now performed additional experiments demonstrating that the efficacy of trastuzumab/FAP-IL2v was abrogated when NK cells were treated with anti-CD16 neutralizing antibodies (supplementary figure 5e). Taking these facts in consideration, and since CD16 constitutes the main receptor triggering ADCC (Capuano *et al*, 2021; Sordo-Bahamonde *et al*, 2020), we conclude that NK cells are key players in the therapeutic effect of FAP-IL2v combined with trastuzumab.

We have included these data and relevant references in the manuscript to address this reviewer concerns.

2. The fragmented and non-consistent nature of the study:

The study is fragmented in two ways:

A. Fibroblasts are those that are supposed to connect between the different parts of the study, but this is actually not the case.

In general, the study has three parts.

- Part 1: Studies of the impact of fibroblasts on tumor cell killing, in the setting of TTZ+ICs.
- Part 2: Studies of IL-2 in advancing the killing effect; the authors have used a FAP-IL2v construct, in which IL-2 was conjugated to antibodies to FAP, a molecule that is typically expressed by CAF-S1 cells.
- Part 3: Studies of TGF, suggesting that TGF induces FAP expression by fibroblasts, that then binds FAP-IL-2v.

Despite the fact that fibroblasts are the linking theme of the study, the experiments of Part 2 did not always control for the effect of fibroblasts. To give an example: The experiments of Figures 4e demonstrated that FAP-IL2v elevated the efficacy of IC-mediated killing of BC cells in the presence of TTZ, in a fibroblast-containing setting. However, it is not clear if FAP-IL2v indeed bound to fibroblasts, leading then to IL-2-induced effects. The control of similar

experiments without fibroblasts should have been performed. The same is correct for the findings described in Figure 5g.

Following this reviewer's request, we have now performed these experiments without fibroblasts. We observed no significant impact of FAP-IL2v on trastuzumab efficiency in absence of BFs indicating that FAP-expressing cells are needed for FAP-IL2v activity. These data are now presented in supplementary figure 3c, supplementary figure 5d and incorporated in the text.

"In the presence of ICs, trastuzumab/FAP-IL2v regimen neither impacted BCCs abundance in a model lacking FAP-expressing BFs when compared to trastuzumab/DP47-IL2v or trastuzumab monotherapy (supplementary figure 3c)."

"NK cells...trastuzumab and FAP-IL2v combination induced a significant reduction of BCCs compared to controls (figure 5g), which could not be achieved in the absence of BFs (supplementary figure 5d)."

The opposite experimental approach was taken by the authors in the in vivo experiments of Figure 5f, where the experiments were performed ONLY in the absence of fibroblasts, thus the important aspect of response in the presence of fibroblasts was missing.

We realize that we did not clearly explain that after cancer cells transplantation in mouse, stromal cells of murine origin (e.g. fibroblasts) are commonly recruited by the nascent tumor. We have performed IHC analysis using FAP antibody (4B9 clone) that confirmed the presence of a murine microenvironment expressing FAP in grown tumors. These data are now presented in supplementary figure 5c and incorporated in the text.

"After cancer cells inoculation, fibroblasts of murine origin were recruited by the nascent tumor. Consequently, macroscopic tumors expressed stromal FAP protein at the time of treatment initiation (supplementary figure 5c)."

Concerning Part 3: Many aspects of the TGF story were not completed and the connection of this part to Parts 1 and 2 is loose.

B. The use of 3 HER2+ BC cell lines, in a non-consistent manner:

The 3D spheroid system has used 3 different human BC cell lines. However, in different sections of the Figures, different cell types were used. Moreover, in Figs. 2a and 2e, Figs. 3e and 3f and Figs 4d and 4e, the type of cell line used was not indicated.

Figure 2c demonstrates results of SKBR3 and HCC1954 cells, obtained in the 3D spheroid system. The next aspect of the 3D spheroid system – in Figures 2e, 2f, 3e and 3f – was performed with one of the cell lines, and it is not indicated which one it was. In the follow up part, in Figures 4d and 4e, it is also not clear which two cell lines were used. The next stage of the study was performed with SKBR3 cells (Figs 5a-5d). Then, the study was continued to the TGF aspects, performed with HCC1954 cells.

Although parts of some of the aspects were analyzed in additional cell lines and were presented in the Supplementary materials, the overall flow of study is inconsistent and hard to follow. It is also very difficult to grasp how the observations of one section relate to other sections.

The manuscript has been revised following this reviewer's request and HCC1954 BCCs related experiments are now generally presented in the main figures whereas equivalent experiments with additional BC cell lines are displayed in supplementary figures. BC cell types are defined in figures and legends. Panels including two BC cell types (*e.g.* HCC1954 and SKBR3) are now presented separately, with HCC1954 BCCs related experiments displayed in main figures (figure 2d and 4f,g) and experiments performed with SKBR3 BCCs shown in supplementary figures 1h and 3b,e.

3. Findings on TGF:

In Part 3 of the study, the authors suggest that TGF induces FAP expression by fibroblasts, that then binds FAP-IL-2v.

If indeed so, it would have been expected that tumor cell killing in 3D spheroids of BC cells + ICs + fibroblasts + FAP-IL2v will be stronger than in 3D spheroids not containing fibroblasts, but this comparison was not performed.

As suggested by this reviewer, the manuscript includes now experiments performed in presence of ICs or NK cells but in absence of fibroblasts. In this setting, trastuzumab/FAP-IL2v regimen did not induce further anti-cancer response against BCCs compared to trastuzumab/DP47-IL2v or trastuzumab monotherapy. These results are complementing our original findings and indicate that FAP-IL2v effectiveness depends on the presence of FAP-expressing cells. These data are now presented in supplementary figure 3c, supplementary figure 5d and are incorporated in the text.

“In the presence of ICs, trastuzumab/FAP-IL2v regimen neither impacted BCCs abundance in a model lacking FAP-expressing BFs when compared to trastuzumab/DP47-IL2v or trastuzumab monotherapy (supplementary figure 3c).”

“NK cells...trastuzumab and FAP-IL2v combination induced a significant reduction of BCCs compared to controls (figure 5g), which could not be achieved in the absence of BFs (supplementary figure 5d).”

Moreover, the authors have not shown directly that FAP is needed for tumor cell killing in this setting, e.g. by knocking-down/out FAP expression in the fibroblasts. The evidence they have provided was indirect through the use TGF inhibition that has led to reduced expression of FAP.

In the original version of this manuscript, we have used a non-targeting antibody fused to IL2v (DP47-IL2v) as a control to address the specific need of FAP targeting antibody (4B9 clone) fused to IL2v (FAP-IL2v). Our data indicated that FAP-IL2v increased trastuzumab effectiveness compared to non-targeting DP47-IL2v in presence of FAP-expressing BFs. Our revised manuscript includes new pieces of data demonstrating the need of FAP-expressing cells for FAP-IL2v efficacy. Above mentioned experiments performed with BCCs monocultures devoid of FAP-expressing cells indicate that FAP-IL2v does not promote anti-cancer activity in this setting (supplementary figure 3c, supplementary figure 5d). To test the specific recognition of BFs by FAP 4B9 antibody fragment of FAP-IL2v, we have now performed flow cytometry assays with BFs, BCCs or ICs. Our results indicate that FAP 4B9 antibody binds specifically to BFs compared to ICs and BCCs (supplementary figure 3d). Altogether, these data suggest that FAP-IL2v binding to BFs enriches the tumor with IL2v, thus potentiating the immune activity initiated by trastuzumab. These data were incorporated in the main text.

“In the absence of ICs, trastuzumab and FAP-IL2v combined therapy did not induce significant cytotoxicity against BCCs when compared to HER2+3DiBC tumors inoculated with trastuzumab and DP47-IL2v (figure 4f, supplementary figure 3b). In the presence of ICs, trastuzumab/FAP-IL2v regimen neither impacted BCCs abundance in a model lacking FAP-expressing BFs when compared to trastuzumab/DP47-IL2v or trastuzumab monotherapy (supplementary figure 3c). Indeed, BFs were specifically recognized by the FAP-targeting antibody fragment of FAP-IL2v (supplementary figure 3d).”

“NK cells...trastuzumab and FAP-IL2v combination induced a significant reduction of BCCs compared to controls (figure 5g), which could not be achieved in the absence of BFs (supplementary figure 5d).”

Additional comments:

1. Title:

The authors have demonstrated mechanisms that control IC-mediated killing of HER2+ BC cells, but they did not reach the point of “predicting” resistance of HER2+ patients to TTZ treatments. This may be a possible implication of the study, but the title should describe the findings and not their possible implications.

Manuscript title was changed as requested by reviewers 1 & 2.

Cancer-associated fibroblasts examination in HER2+ breast cancer reveals a new approach enhancing immunity to overcome resistance to therapy

2. IHC patient analysis of Figure 1:

CD45 expression signifies many immune cell types that may have opposing effects on malignancy. To connect patient IHC data to the study, analysis of NK cells should have been performed and presented in Part 2 of the study.

As requested by this reviewer, we have now performed IHC analysis using CD56 antibody as surrogate marker for NK cells. Our data indicate that the presence of CD56 positive cells was significantly increased in tumors from patients better responding to treatment. Corresponding data are displayed in figure 3e and supplementary figure 2b. Manuscript was updated accordingly.

“IHC analyses in our 22 cases of HER2+ BC confirmed that tumors from patients that remained disease-free following trastuzumab treatment showed an increased number of intra-tumoral CD56+ immune cells (NK cells) compared to tumors from relapsing patients (figure 3e, supplementary figure 2b).”

3. The immune gene signature of Figure 1: The signature contains genes with opposing roles in regulation of cancer progression. Did the authors try to dissect the GSEA analysis into subgroups of genes, in effort to achieve some kind of dissociation between genes that have been typically classified as malignancy-supporting genes from genes that are typically identified with anti-malignancy functions?

We thank this reviewer for raising this concern. As requested, a list of enrichment scores related to each immune gene has been added as supplementary table 4 for the two analyzed cohorts. In contrast with the full immune signature, single genes were in general not significantly upregulated in responders to treatment vs non-responders. Of note, IL2 receptor subunit beta (IL2RB) was one of the genes significantly enriched in responders from both cohorts.

However, general immune and stromal signatures used in this study for GSEA were originally assembled by Yoshihara and colleagues as a tool (Estimation of STromal and Immune cells in MAlignant Tumors using Expression data; ESTIMATE) to infer the presence of infiltrating stromal/immune cells in tumor tissues using gene expression data (Yoshihara *et al*, 2013). We agree that GSEA analysis was somewhat not the most relevant analysis. In this context and in order to answer Reviewer 2 comment, we have now performed ESTIMATE analyses instead of GSEA (figure 1b). Our results indicate that poor-prognosis/non-responder patients are characterized by increased stromal and decreased immune scores. Manuscript was updated accordingly.

“Next, we performed ESTIMATE (Estimation of STromal and Immune cells in MAlignant Tumors using Expression data) analyses (Yoshihara et al, 2013) to infer the presence of infiltrating stromal and immune cells in two independent cohorts of HER2+ BC patients treated with trastuzumab. In a first cohort of 63 patients responsive or resistant to treatment (Prat et al, 2014), immune/stroma enrichment score ratio was significantly upregulated in biopsies collected before treatment from patients achieving pathological complete response (pCR) (figure 1b, left panel). In a second cohort of 51 HER2+ BC cases (Triulzi et al, 2015), ESTIMATE analysis indicated that non-relapsing patients after adjuvant HER2-targeted treatment were also characterized by increased immune/stroma ratio (figure 1b, right panel).”

4. Figure 2c, Figures 4d and 4e:

The data of two different cell lines were combined in one statistical analysis. This is a non-conventional mode of analysis. Each cell line should be analyzed alone (HCC1954 separately from SKBR3).

As mentioned above, panels including two BC cell types (HCC1954 and SKBR3) are now displayed separately, with HCC1954 related experiments in main figures (figure 2d and 4f,g). Equivalent experiments performed with SKBR3 are presented in corresponding supplementary figures 1h and 3b,e. Statistical analyses are now performed separately for HCC1954 and SKBR3.

5. Figure 3c:

What is the number of patients in each group of the cohort?

Results are now provided for 22 HER2+ breast cancer patients treated with anti-HER2 therapy (13 responsive and 9 non-responsive to treatment). This data has been added to figure 1a, 3e, 4a and to methods section.

How did the authors determine NK-IL2RS high vs. NK-IL2Rs low?

NK-IL2RS score discriminating NK-IL2RS high and NK-IL2Rs low patients was determined using receiver operating characteristic (ROC) curve analysis as described elsewhere (Budczies *et al*, 2012). This information was added to methods section.

6. Figure 4a:

The Figure demonstrates FAP expression in HER2+ BC patients vs normal breast. It is not tightly connected to the study because it does not present data connected to TTZ treatment.

How many patient samples were analyzed by IHC for FAP expression?

In agreement with this reviewer, we have now replaced this panel with IHC analysis of FAP protein expression in tumors from 22 patients with HER2+ BC responsive or non-responsive to trastuzumab (figure 4a). Our data indicate that FAP high tumors are less responsive to treatment. IHC staining illustrating low and high FAP protein expression in HER2+ BC tumors are displayed in figure 4b. Manuscript was updated accordingly.

"IHC analysis performed in BC patients showed that increased FAP protein expression was detected in the TME of tumors unresponsive to trastuzumab (figure 4a,b)."

7. Figure 5a:

Control of DP47-IL2v is missing.

Controls were added to figure 5a as requested. Data indicate that DP47-IL2v treatment does not associate with CD45 and NCR1 upregulation in our *ex vivo* model.

8. Figure 5g (and 5f):

The figure was supposed to demonstrate the activities of NK cells, but the text inside the figure states "ICs" and not "NK". The legend to Figure 5g does not clarify this inconsistency, because the description of part g is missing. The legend of part f is also missing.

We thank the reviewers for pointing out this mistake. Figure 5f, 5g and corresponding legends have been corrected.

“(f) Growth kinetics of macroscopic tumors derived from subcutaneous injection of HCC1954 cells into nude mice treated with DP47-IL2v (n=11), FAP-IL2v (n=12), TTZ (n=8), TTZ/DP47-IL2v (n=12), TTZ/FAP-IL2v (n=11) compared to control (IgG1; n=7). Values are mean \pm s.e.m. (g) Mean bioluminescent tracking of BCCs (HCC1954) in HER2+3DiBCs treated as indicated in presence of NK cells. IgG1 was used as control for TTZ. Values are mean \pm s.e.m.”

9. The fibroblast lines BF1 and BF2:

Were they CAFs or NAFs? How were they validated? How were they immortalized?

BFs were kindly provided by P. Gascon/P. Bragado laboratory (IDIBAPS, Barcelona) and are described elsewhere (Fernández-Nogueira *et al*, 2020). In brief, BFs were expanded in culture from digested pieces of fresh HER2+ BC tissue. Differential trypsinization was performed to further isolate the fibroblast population. A retroviral vector pMIG (MSCV-IRES-GFP) expressing hTERT and GFP was used to immortalize and label BFs. BFs were characterized by measuring the expression of α -SMA (alpha smooth muscle actin) (Fernández-Nogueira *et al*, 2020) and FAP (figure 4e and supplementary figure 3a). Methods section was updated accordingly.

10. The term “infiltration”:

The term “infiltration” was often used by the authors to describe the increased presence of immune cells in the 3D spheroid system, in the presence of IL-2. However, IL-2 could have led to elevated proliferation of activated T cells and of NK cells. Thus, the term should be changed to “infiltration/proliferation”.

As requested, the term infiltration was replaced by infiltration/expansion or by abundance.

Reviewer #2 (Remarks to the Author):

In this study, the authors study report the development of a fully humanized immunocompetent ex vivo method of HER2+ breast cancer cells, cancer-associated fibroblasts (CAFs) and immune cells that recapitulates patients’ response to trastuzumab. The authors further identify distinct stromal traits predicting the lack of benefit from trastuzumab therapy. Notably, stroma-targeted stimulation of IL-2 pathway with Simlukafusp Alfa, an immunocytokine composed of IL2v fused to a FAP-targeting antibody (FAP-IL2v) currently in clinical trial, restored trastuzumab activity in unresponsive tumors.

This study is well-conducted, well-controlled and data for the most part support the conclusions. The study is also very timely and will certainly stimulate broad interest in the field on immuno-oncology, especially considering the renewed interest in IL-2: currently at least 11 candidate drugs based on engineered IL-2 are currently at IND stage or in clinical trials (Nat Rev Drug Discovery 20, 163-165; 2021).

Below are my specific comments:

1. The abstract should mention that the immunocompetent model used is an ex vivo model.

We fully agree with this reviewer. The abstract was modified accordingly.

“we developed a fully humanized immunocompetent model of HER2+ BC recapitulating ex vivo biological processes that associate with patients’ response to treatment”

2. Figure 1: Did you use the ESTIMATE method from ref 43?

In the original version of this manuscript, immune and stromal signatures from Yoshihara and colleagues (Yoshihara *et al*, 2013) were used for GSEA analyses only. In order to address this reviewer concern, revised manuscript now includes ESTIMATE analyses instead of GSEA and shows immune/stromal enrichment ratios association with clinical data (figure 1b). Our results indicate that poor-prognosis/non-responder patients are characterized by increased stromal and decreased immune scores. Manuscript was updated accordingly.

*“Next, we performed ESTIMATE (Estimation of STromal and Immune cells in MAlignant Tumors using Expression data) analyses (Yoshihara *et al*, 2013) to infer the presence of infiltrating stromal and immune cells in two independent cohorts of HER2+ BC patients treated with trastuzumab. In a first cohort of 63 patients responsive or resistant to treatment (Prat *et al*, 2014), immune/stroma enrichment score ratio was significantly upregulated in biopsies collected before treatment from patients achieving pathological complete response (pCR) (figure 1b, left panel). In a second cohort of 51 HER2+ BC cases (Triulzi *et al*, 2015), ESTIMATE analysis indicated that non-relapsing patients after adjuvant HER2-targeted treatment were also characterized by increased immune/stroma ratio (figure 1b, right panel).”*

3. Figure 2b: Which factor(s) expressed by BFs is (are) responsible for increased BCCs proliferation.

Fibroblasts from the tumor microenvironment secrete a plethora of factors that intervene in processes such as cancer cells proliferation, invasion and metastasis (Sahai *et al*, 2020; Kalluri,

2016). Some of these factors include IL-6, IL-8, IL-11, POSTN, FGF2/7, PDGF, HGF and IGF-2 among others (Elwakeel *et al*, 2021; Wu *et al*, 2021; Linares *et al*, 2021). This information has been incorporated in the discussion.

Please show correlation (in supplementary) between luciferase expression and cell proliferation/cell death, as this is the main read-out.

As requested, the correlation between cancer cells abundance and bioluminescence activity has been assessed by luciferase assay applied to serial dilutions of luciferase-expressing BCCs. As shown in supplementary figure 1c, doubling the amount of BCCs corresponds to a doubling of the measured bioluminescence. Manuscript was updated accordingly.

4. Please add methodological details regarding the plasmid used to engineer bioluminescent/fluorescent cancer cells and how the luciferase assay was performed.

Methods section was modified in accordance with this reviewer comment.

“fusion protein reporter construct encoding mCherry and firefly luciferase was cloned into FUW lentiviral vector (Lois et al, 2002) under control of CMV promoter (Calon et al, 2012). Stable expression of luciferase and mCherry was obtained upon lentiviral infection of cancer cells followed by antibiotic selection (Zeocin; InvivoGen, Toulouse, France). Homogenous expression of the reporter was assessed by measuring mCherry fluorescence in infected cells sorted by FACS. TGF-beta activity reporter cells were generated after infection with TGF/SMAD Luciferase Reporter Lentivirus (Kerafast) followed by antibiotic selection (Puromycin; InvivoGen, Toulouse, France). Luciferase reporter assays were performed by adding luciferin potassium salt (Resem BV, Lijnden, Netherlands) at a final concentration of 50ug/ml to cells cultured in 96-well flat bottom clear, white walls plates (Corning Incorporated Life Science, Kennebunk, USA). Bioluminescence was measured following luciferin injection in each well with Microplate Luminometer, Orion II (Berthold Detection Systems, Pforzheim, Germany) and Simplicity 4.2 software.”

5. Figure 2d: Please discuss how immune cells (IC) from naive patient may increase BCCs' proliferation.

Immune cells can contribute to tumor growth through the establishment of a beneficial inflammatory niche. In this line, pro-tumorigenic inflammation has become one of the hallmarks of cancer (Hanahan & Weinberg, 2011), as numerous studies have shown that tumor development depends on interactions between cancer cells and inflammatory cells through

secretion of cytokines. For instance, IL-1 β and IL-18 secreted in response to cellular stress has been shown to trigger tumor progression through activation of NF- κ B (Van Gorp & Lamkanfi, 2019). Other cytokines such as TNF- α , IL-6, IL-17 and IL-23 may lead to tumor growth through activation of inflammatory pathways such as NF- κ B and STAT3 (Liubomirski *et al*, 2019; Zamarron & Chen, 2011; Chabab *et al*, 2020). We have now included this information in our discussion to answer this reviewer's concern.

6. Figure 2c-f: Please indicate the ratio of BCC/BFs used in the figure legends.

A 1/5 (BCCs/BFs) ratio was used in our experimental setting. As requested, figure legend and methods section were modified to include this information.

Is the ratio of BCC/BFs used in all experiments representative of the physiological ratio found in primary tumors and/or metastatic lesions?

We thank this reviewer to raise this important concern. The stroma fraction is highly variable across breast cancer patients and may range from approximately 5 to 95% of the tumor (Micke *et al*, 2021). Of note, Tumor/Stroma Ratio (TSR) assessing the amount of tumor-associated stroma has been extensively described in breast cancer (Kramer *et al*, 2019). With 80% of fibroblasts, our HER2+3DiBC model may be categorized as stroma-high breast tumor by TSR evaluation. This information has been incorporated in the discussion.

"With 80% of BFs and 20% of BCCs, this HER2+3DiBC model is comparable to stroma-high breast tumors (Kramer et al, 2019)."

7. Can you assess the impact of BCC/BFs ratio on TTZ efficacy and ICs infiltration?

Following this reviewer's request, we have now measured the impact of BCCs/BFs ratio on trastuzumab efficiency in presence of ICs. Our data indicate that increasing abundance of BFs in HER2+3DiBC reduced trastuzumab-induced immune response against BCCs in a dose dependent manner (supplementary figure 1i). Manuscript was updated accordingly.

Are ICs excluded from the spheroids because the number of fibroblasts is too important (i.e. acts as a physical barrier)?

We have performed IHC analyses on FFPE sections of HER2+3DiBC to address this reviewer concern. FAP immunostaining indicated that BFs are expanding in a loose fashion leaving extended acellular spaces in the 3D matrix (figure 4e). Together with the immunofluorescence studies performed in HER2+3DiBC (figure 2a and 4h), this observation indicates that BFs are

not self-organizing into a physical barrier, thus suggesting alternative mechanisms such as the production of factors impeding ICs entry in HER2+3DiBC. This information has been now incorporated in the manuscript.

8. Figure 3 conclusion (line 170-173): please clarify how data implies 'the existence of specific stromal signaling associated with resistance to anti-HER2 mAbs'. This could be made more clear in the text.

The conclusion was modified according to this reviewer's suggestion as follow.

"Thus, our findings underscore that stromal characteristics -increased CAF-S1 and pCAF, decreased NK-IL2RS- associate with resistance to anti-HER2 mAbs."

9. Line 184: "thus confirming our observations in patients". It is not clear which observations in patients you are referring to.

The sentence was modified as follow according to this reviewer's comment.

"Trastuzumab and rhIL2 combination strongly enhanced immune infiltration/expansion leading to an 80% reduction of BCCs without significant effect over BFs (figure 3f), which coincides with the association between IL2 activity in NK cells and trastuzumab effectiveness observed in patients."

10. Figure 5: Please explain how you discriminated between the immune infiltrate inside of the spheroids vs immune cells excluded from the spheroids, for qPCR or FACS analysis? If you are not measuring immune cells infiltration per se, then the text should be modified accordingly.

To specifically study the immune infiltrate in HER2+3DiBC tumors, non-infiltrating immune cells were excluded prior to analyses as follow. Three PBS wash were performed following culture media retrieval. BME drops containing the immune infiltrate were then pulled out from the culture well with a cut-off P1000 pipet tip and processed for RNA extraction or cell sorting analyses. Methods section was modified to clarify this point.

11. Figure 5a-b: Can you comment on the very interesting difference between FAP-IL2v and DP47-IL2v? Cytotoxic activity is improved when the IL2 is targeting the fibroblasts, even within this in vitro setting. In figure 4a, FAP-IL2v increased CD45+ cells "infiltration" (or expansion), can you assess whether DP47-IL2v equally increased CD45+ mRNA level within the spheroids or this is dependent on the fibroblast targeting of IL2?

As requested, we have now performed this experiment with untargeted DP47-IL2v and assessed its capacity to modulate CD45+ mRNA level within the spheroids (figure 5a). We observed that DP47-IL2v treatment did not associate with CD45 and NCR1 upregulation. This finding suggests that CD45+ cells abundance in HER2+3DiBC depends on the capacity of FAP-IL2v to bind to FAP-expressing cells. Of note, our experimental setup *ex vivo* corresponds to a 4-hours treatment of HER2+3DiBC with FAP-IL2v or with DP47-IL2v followed by unbound compounds retrieval upon PBS wash steps. In this setting, FAP-IL2v that bound to BFs was sufficient to potentiate trastuzumab-dependent ADCC against BCCs. Manuscript was updated accordingly.

12. Figure 5c-d: The frequency of other immune cell types (CD4, CD8, monocytes, B cells) should be represented (in supplementary).

As requested by this reviewer, corresponding data are now represented in supplementary figure 4f and supplementary table 5. We observed that CD4+ and CD8+ T cells as well as CD20+ B cells abundance was not significantly modulated by trastuzumab and FAP-IL2v regimen in our experimental setting. Of note, CD14+ monocytes abundance was in some cases rather decreased upon treatment. These findings further underscore the importance of NK cells for treatment effectiveness in our experimental setting. Manuscript was updated accordingly.

13. Figure 5f-g: Legend is missing/incorrect. Panel g: were purified NK cells used? If so, it should be indicated in the figure legends.

We thank the reviewers for pointing out this mistake. Figure 5g, 5f and corresponding legends have been corrected.

“(f) Growth kinetics of macroscopic tumors derived from subcutaneous injection of HCC1954 cells into nude mice treated with DP47-IL2v (n=11), FAP-IL2v (n=12), TTZ (n=8), TTZ/DP47-IL2v (n=12), TTZ/FAP-IL2v (n=11) compared to control (IgG1; n=7). Values are mean ± s.e.m. (g) Mean bioluminescent tracking of BCCs (HCC1954) in HER2+3DiBCs treated as indicated in presence of NK cells. IgG1 was used as control for TTZ. Values are mean ± s.e.m.”

14. Line 533-534 (Methods). What clinical alterations did you observe that led to mouse exclusion from the study? Did mouse exclusion lead to data exclusion? The total number of mice per group and number of mice excluded should be specified.

Cancer cells were inoculated subcutaneously in both flanks of 5–6 weeks-old swiss nude mice. One mouse showing significant weight loss, abnormal posture and lack of mobility after cancer

cells inoculation was excluded from the study. Another mouse didn't develop tumors in either flank and was also excluded. In both cases, exclusion occurred prior to treatment initiation and did not lead to data exclusion. Mice bearing macroscopic tumors were injected with either control IgG1 (4 mice; 7 initiated tumors prior to treatment), trastuzumab (4 mice; 8 initiated tumors prior to treatment), FAP-IL2v (6 mice; 12 initiated tumors prior to treatment), DP47-IL2v (6 mice; 11 initiated tumors prior to treatment), trastuzumab/DP47-IL2v (6 mice; 12 initiated tumors prior to treatment) or trastuzumab/FAP-IL2v (6 mice; 11 initiated tumors prior to treatment). Tumor expansion was assessed on each flank by caliper measurements. Methods section was modified accordingly.

15. Figure 6: Where is the TGF- β coming from in your ex vivo model and in HER2+ cancer. For the moment, the production of TGF- β by BCCs is only presumed.

TGF- β can be secreted by a myriad of cell types in tumors (Massagué, 2008; Drabsch & Ten Dijke, 2011; Seoane & Gomis, 2017). In particular, fibroblasts produce elevated levels of TGF- β in the tumor microenvironment (Calon *et al*, 2014; Isella *et al*, 2015; Kalluri, 2016). Similarly, fibroblasts are the main producers of TGF- β in our experimental setting (supplementary figure 6e). However, TGF- β is commonly secreted as a latent complex that needs to be activated. We observed that TGF- β activity was increased in BFs only when co-cultured with BCCs (figure 6e, supplementary figure 6f). These findings suggest that BCCs may be required to process latent TGF- β into its active form. Data were incorporated in Results section and Discussion was updated accordingly.

16. Can you comment on the possible effect of FAP-IL2v on CD8+T cell activity?

As observed in many cases of breast cancer, our model recreates features of immune evasion that may include T cells exclusion which is likely due to CAF-associated factors (Salmon *et al*, 2019; Costa *et al*, 2018). T cells were extensively studied in the context of anti-cancer immunity and their presence in the tumor has been associated with better outcome including in HER2+ BC (Capietto *et al*, 2011; Stanton & Disis, 2016). However, CD4+ and CD8+ T cells expansion/infiltration was not significantly modulated by trastuzumab and FAP-IL2v regimen in our experimental setting (supplementary figure 4f). In contrast, NK cells were constantly increased within the immune infiltrate of treated HER2+3DiBC (figure 5c, supplementary figure 5a), thus suggesting a prominent role of NK cells in driving trastuzumab/FAP-IL2v therapeutic efficacy.

We have now included this information and related references in the main text to address this reviewer's concern.

17. In the discussion, it would be appropriate to briefly mention the other engineered IL-2 variants currently in clinical trials (Bempegaldesleukin, Nemvaleukin alfa, SAR444245, RG6279, CUE-101).

Discussion was modified according to this reviewer's comment.

"IL2 has drawn increasing attention in oncology (Mullard, 2021; Mani et al, 2009) and several IL2-based therapies (Diab et al, 2020; Ptacin et al, 2021; Quayle et al, 2020) are currently evaluated in the clinical setting (e.g. NCT02983045, NCT04009681, NCT04303858, NCT03978689). Our findings derived from modelling HER2+ BC ex vivo shed new light on the capacity of IL2 to enhance trastuzumab effectiveness against BCCs."

18. Consider changing the title to better reflect findings. 'Tumor microenvironment examination...' is quite vague.

Manuscript title was modified as requested by reviewers 1 & 2.

Cancer-associated fibroblasts examination in HER2+ breast cancer reveals new approach enhancing immunity to overcome resistance to therapy

Reviewer #3 (Remarks to the Author):

About 50% of HER2+ breast cancer (BC) patients do not benefit from HER2-targeted therapy and almost 20% of them would relapse after treatment. This study identified distinct stromal traits predicting the lack of benefit from therapy. The authors observed that TGF- β -activated CAF, specifically, CAF-S1 subpopulation were increased in tumors resistant to therapy, and associated with low activity of IL2. Stroma-targeted stimulation of IL2 pathway in unresponsive tumors restored trastuzumab anti-cancer efficiency, and FAP-IL2v may restore trastuzumab-mediated ADCC against BCCs in unresponsive HER2+ BC. The topic of the manuscript is interesting and has clinical significance.

Effect of IL2 in combination of trastuzumab in MBC was contractively reported by different studies. Results from phase II trial of trastuzumab in combination with low-dose IL2 in patients with metastatic breast cancer who have previously failed trastuzumab did not find clinical benefit and lack of expansion of NK cells in response the treatment (Mani et al. Breast Cancer Res Treat. 2009 Sep; 117(1): 83–89). Currently, FAP-IL2v is in Phase 1b/2 clinical development

as a monotherapy and in combination with trastuzumab and cetuximab. The finding from this study could advance the understanding how FAP-IL2v functional enhance the therapeutic effect of Trastuzumab, and which would have better benefit to the treatment.

Overall, the manuscript is prepared well, and the approaches used in the study are appropriate. The mouse model they developed for the study is novel. The data presented is clear with good quality figures.

We thank this reviewer for his/her comments. Reference to Mani et al. (Breast Cancer Res Treat. 2009) was included in the discussion.

Reviewer #4 (Remarks to the Author):

The authors of this manuscript propose that resistance to trastuzumab in HER2 breast cancer may be mediated by stromal cells, in particular CAF-S1 cells activated by TGF β and that this resistance can be reversed by TGF β inhibitors. They also attempt to show that NK cells are important in response to trastuzumab that may be mediated by IL-2 activity. The following comments and suggestions are provided:

1. The authors show that naive ICs are somehow stimulatory to tumor growth. Is there evidence for this in the literature?

Immune cells can contribute to tumor growth through the establishment of a beneficial inflammatory niche. In this line, pro-tumorigenic inflammation has become one of the hallmarks of cancer (Hanahan & Weinberg, 2011), as numerous studies have shown that tumor development depends on interactions between cancer cells and inflammatory cells through secretion of cytokines. For instance, IL-1 β and IL-18 secreted in response to cellular stress has been shown to trigger tumor progression through activation of NF- κ B (Van Gorp & Lamkanfi, 2019). Other cytokines such as TNF- α , IL-6, IL-17 and IL-23 may lead to tumor growth through activation of inflammatory pathways such as NF- κ B and STAT3 (Liubomirski *et al*, 2019; Zamarron & Chen, 2011; Chabab *et al*, 2020). We have now included this information in our discussion to answer this reviewer's concern.

2. In figure 1d the authors show that a stromal gene signature is associated with resistance to trastuzumab in 2 clinical datasets. Why is it then necessary to invoke the CAF-S1 subset when a generalized stromal signature and unsorted fibroblasts that are mixed with BC are shown to produce this effect? Were other CAF subtypes examined?

We thank this reviewer to raise this important concern. The stromal signature used in this manuscript for GSEA was originally assembled by Yoshihara and colleagues to create a companion tool for the Estimation of STromal and Immune cells in MAlignant Tumors using Expression data (ESTIMATE) (Yoshihara *et al*, 2013). ESTIMATE is relevant to infer the presence of infiltrating stromal/immune cells in tumor tissues using gene expression data. However, this general stromal signature rather meant for quantitative purpose does not associate with specific physiological processes. In agreement with Reviewer 2, we have now performed ESTIMATE analyses instead of GSEA (figure 1b). Our results indicate that poor-prognosis/non-responder patients are characterized by increased stromal and decreased immune scores.

In contrast to above mentioned stromal signature, CAF-S1 gene set associates with a subset of immunosuppressive fibroblasts in breast cancer, thus providing information about the quality of the tumor stroma. We shared this reviewer's concern and wondered whether any CAF signature may be enriched in the TME of patients unresponsive to treatment. To address this matter, we have now analyzed two additional CAFs populations -pCAFs and sCAFs- that were recently described in breast cancer by Friedman and colleagues (Friedman *et al*, 2020). In this study, authors show that pCAFs also associate with immunosuppression. Conversely, authors suggest that the presence of sCAFs may activate the immune system and improve clinical outcome in breast cancer. Remarkably, CAF-S1 and pCAFs subsets are both FAP-expressing fibroblasts found close to cancer cells. On the other hand, sCAFs are sharing with CAF-S3 -a subset that was not molecularly analyzed- increased S100A4 expression and localization away from cancer cells (Friedman *et al*, 2020). Similar to CAF-S1 analysis, GSEA indicated that pCAFs gene set was enriched in patients non-responding to treatment. In contrast, sCAFs was either significantly enriched in responders of the first cohort or had equivalent expression in pCR and RD subsets of the second cohort. These new data showing differential expression of CAF-associated signatures were incorporated in the main text and in figure 1c-e.

3. The authors indicate that NKIL2RS explains about 70% of cancer relapses after HER2-targeted therapy independent of the main clinical variables, however, this is not consistent with the data nor can this sort of extrapolation be made. Also in the same table, it is unclear why the difference between low (.784) and high (6.336) grade is not significant whereas the difference between low .096 and high (.606) NKIL2RS is highly significant.

As requested, conclusions from figure 3c were modified in the main text as follow.

“Consequently, low NK-IL2RS expression was an effective predictor of recurring disease after treatment (figure 3c), independently of the main clinical variables (figure 3d).”

Also, we realize that we did not explain clearly the piece of data displayed in figure 3d. This table displays the multivariate analysis performed when considering all variables. HR (hazard ratio) values correspond to the exponentiation of the odd ratios (risk estimate) for each variable. Of note, a value below 1 corresponds to a decreased risk of relapse (e.g. 0.242 for NKIL2RS) and a value above 1 indicates an increased risk of relapse (e.g. 2.229 for GRADE). A 95% confidence interval (CI) is displayed for each hazard ratio (e.g. 0.784 to 6.336 for GRADE and 0.096 to 0.606 for NKIL2RS). For the GRADE variable, a confidence interval including an upper value superior to 1 (increased risk) and a lower value inferior to 1 (decreased risk) associates with a non-statistically significant hazard ratio. In contrast, a confidence interval including both upper and lower value inferior to 1 has a good chance to associate with a statistically significant hazard ratio for a predictor of decreased risk of relapse (e.g. NKIL2RS variable). The methods section has been updated accordingly.

4. On page 6, paragraph 2 it is indicated that the FAP protein is associated with expression of CAF-S1 and inversely associated with NKIL2RS yet FAP is not associated with response in the two clinical cohorts. This seems counterintuitive is there an explanation for this?

We fully agree with this reviewer's comment. One explanation may reside in the fact that FAP is represented by a single probe in both Affymetrix and Illumina transcriptomic datasets. This lack of redundancy -several probes targeting the same transcripts- could limit the detection of alternatively spliced transcript variants and decrease the strength of the analysis. However, we have now performed IHC analysis of FAP protein expression in 22 patients responsive or non-responsive to trastuzumab included in our study (figure 4a). Despite the reduced size of this cohort, data suggest that tumors with increased FAP expression are less responsive to treatment. IHC illustrating low and high FAP expression in HER2+ BC tumors are displayed in figure 4b. Manuscript was updated accordingly.

5. There is no explanation for Fig 5f+g in the legend. Please clarify.

We thank the reviewers for pointing out this mistake. Figure 5g, 5f and corresponding legends have been corrected.

“(f) Growth kinetics of macroscopic tumors derived from subcutaneous injection of HCC1954 cells into nude mice treated with DP47-IL2v (n=11), FAP-IL2v (n=12), TTZ (n=8), TTZ/DP47-IL2v (n=12), TTZ/FAP-IL2v (n=11) compared to control (IgG1; n=7). Values are mean ± s.e.m. (g)

Mean bioluminescent tracking of BCCs (HCC1954) in HER2+3DiBCs treated as indicated in presence of NK cells. IgG1 was used as control for TTZ. Values are mean ± s.e.m.”

6. Under methods please describe how fibroblasts were isolated and characterized.

Methods section has been updated according to this reviewer’s comment.

“BFs were kindly provided by P. Gascon/P. Bragado laboratory (IDIBAPS, Barcelona) and are described elsewhere (Fernández-Nogueira et al, 2020). In brief, BFs were expanded in culture from digested pieces of fresh HER2+ BC tissue. Differential trypsinization was performed to further isolate the fibroblast population. A retroviral vector pMIG (MSCV-IRES-GFP) expressing hTERT and GFP was used to immortalize and label BFs. BFs were characterized by measuring the expression of α -SMA (alpha smooth muscle actin) (Fernández-Nogueira et al, 2020) and FAP (figure 4e and supplementary figure 3a).”

Budczies J, Klauschen F, Sinn B V, Gyo “ Rffy B & Schmitt WD (2012) Cutoff Finder: A Comprehensive and Straightforward Web Application Enabling Rapid Biomarker Cutoff Optimization. *PLoS One* 7: 51862

Calon A, Espinet E, Palomo-Ponce S, Tauriello D V, Iglesias M, Cespedes M V, Sevillano M, Nadal C, Jung P, Zhang XH, et al (2012) Dependency of colorectal cancer on a TGF-beta-driven program in stromal cells for metastasis initiation. *Cancer Cell* 22: 571–584

Calon A, Tauriello DVF & Batlle E (2014) TGF-beta in CAF-mediated tumor growth and metastasis. *Semin Cancer Biol* 25: 15–22

Capietto A-H, Martinet L & Fournié J-J (2011) Stimulated $\gamma\delta$ T Cells Increase the In Vivo Efficacy of Trastuzumab in HER-2+ Breast Cancer. *J Immunol* 187: 1031–1038

Capuano C, Pighi C, Battella S, De Federicis D, Galandrini R & Palmieri G (2021) Harnessing CD16-Mediated NK Cell Functions to Enhance Therapeutic Efficacy of Tumor-Targeting mAbs. *Cancers* 2021, Vol 13, Page 2500 13: 2500

Chabab G, Barjon C, Bonnefoy N & Lafont V (2020) Pro-tumor $\gamma\delta$ T Cells in Human Cancer: Polarization, Mechanisms of Action, and Implications for Therapy. *Front Immunol* 11: 2186

Costa A, Kieffer Y, Scholer-Dahirel A, Pelon F, Bourachot B, Cardon M, Sirven P, Magagna I, Fuhrmann L, Bernard C, et al (2018) Fibroblast Heterogeneity and Immunosuppressive Environment in Human Breast Cancer. *Cancer Cell* 33: 463–479.e10

Diab A, Tannir NM, Bentebibel SE, Hwu P, Papadimitrakopoulou V, Haymaker C, Kluger HM, Gettinger SN, Sznol M, Tykodi SS, et al (2020) Bempegaldesleukin (NKTR-214) plus Nivolumab in Patients with Advanced Solid Tumors: Phase I Dose-Escalation Study of Safety, Efficacy, and Immune Activation (PIVOT-02). *Cancer Discov* 10: 1158–1173

Drabsch Y & Ten Dijke P (2011) TGF- β Signaling in Breast Cancer Cell Invasion and Bone

Metastasis. *J Mammary Gland Biol Neoplasia* 16: 97

- Elwakeel E, Weigert A, Baldari CT & Sozzani S (2021) Breast Cancer CAFs: Spectrum of Phenotypes and Promising Targeting Avenues. *Int J Mol Sci* 2021, Vol 22, Page 11636 22: 11636
- Fernández-Nogueira P, Mancino M, Fuster G, López-Plana A, Jauregui P, Almendro V, Enreig E, Menéndez S, Rojo F, Noguera-Castells A, *et al* (2020) Tumor-Associated Fibroblasts Promote HER2-Targeted Therapy Resistance through FGFR2 Activation. *Clin Cancer Res* 26: 1432–1448
- Friedman G, Levi-Galibov O, David E, Bornstein C, Giladi A, Dadiani M, Mayo A, Halperin C, Pevsner-Fischer M, Lavon H, *et al* (2020) Cancer-associated fibroblast compositions change with breast cancer progression linking the ratio of S100A4+ and PDPN+ CAFs to clinical outcome. *Nat Cancer* 2020 17 1: 692–708
- Van Gorp H & Lamkanfi M (2019) The emerging roles of inflammasome-dependent cytokines in cancer development. *EMBO Rep* 20
- Hanahan D & Weinberg RA (2011) Hallmarks of cancer: the next generation. *Cell* 144: 646–674
- Isella C, Terrasi A, Bellomo SE, Petti C, Galatola G, Muratore A, Mellano A, Senetta R, Cassenti A, Sonetto C, *et al* (2015) Stromal contribution to the colorectal cancer transcriptome. *Nat Genet* 47: 312–319
- Kalluri R (2016) The biology and function of fibroblasts in cancer. *Nat Rev Cancer* 16: 582–598 doi:10.1038/nrc.2016.73 [PREPRINT]
- Kramer CJH, Vangangelt KMH, van Pelt GW, Dekker TJA, Tollenaar RAEM & Mesker WE (2019) The prognostic value of tumour–stroma ratio in primary breast cancer with special attention to triple-negative tumours: a review. *Breast Cancer Res Treat* 173: 55
- Linares J, Marín-Jiménez JA, Badia-Ramentol J & Calon A (2021) Determinants and Functions of CAFs Secretome During Cancer Progression and Therapy. *Front Cell Dev Biol* 8: 621070 doi:10.3389/fcell.2020.621070 [PREPRINT]
- Liubomirski Y, Lerrer S, Meshel T, Rubinstein-Achiasaf L, Morein D, Wiemann S, Körner C & Ben-Baruch A (2019) Tumor-stroma-inflammation networks promote pro-metastatic chemokines and aggressiveness characteristics in triple-negative breast cancer. *Front Immunol* 10: 757
- Lois C, Hong EJ, Pease S, Brown EJ & Baltimore D (2002) Germline transmission and tissue-specific expression of transgenes delivered by lentiviral vectors. *Science (80-)* 295: 868–872
- Mani A, Roda J, Young D, Caligiuri MA, Fleming GF, Kaufman P, Brufsky A, Ottman S, Carson WE & Shapiro CL (2009) A phase II trial of trastuzumab in combination with low-dose interleukin-2 (IL-2) in patients (PTS) with metastatic breast cancer (MBC) who have previously failed trastuzumab. *Breast Cancer Res Treat* 117: 83
- Massagué J (2008) TGF β in Cancer. *Cell* 134: 215–230 doi:10.1016/j.cell.2008.07.001 [PREPRINT]
- Micke P, Strell C, Mattsson J, Martín-Bernabé A, Brunnström H, Huvila J, Sund M, Wärnberg F, Ponten F, Glimelius B, *et al* (2021) The prognostic impact of the tumour stroma fraction:

- A machine learning-based analysis in 16 human solid tumour types. *EBioMedicine* 65: 103269
- Mullard A (2021) Restoring IL-2 to its cancer immunotherapy glory. *Nat Rev Drug Discov* 20: 163–165
- Nigro C Lo, Macagno M, Sangiolo D, Bertolaccini L, Aglietta M & Merlano MC (2019) NK-mediated antibody-dependent cell-mediated cytotoxicity in solid tumors: biological evidence and clinical perspectives. *Ann Transl Med* 7: 105–105
- Prat A, Bianchini G, Thomas M, Belousov A, Cheang MCU, Koehler A, Gómez P, Semiglazov V, Eiermann W, Tjulandin S, *et al* (2014) Research-Based PAM50 subtype predictor identifies higher responses and improved survival outcomes in HER2- Positive breast cancer in the NOAH Study. *Clin Cancer Res* 20: 511–521
- Ptacin JL, Caffaro CE, Ma L, San Jose Gall KM, Aerni HR, Acuff N V., Herman RW, Pavlova Y, Pena MJ, Chen DB, *et al* (2021) An engineered IL-2 reprogrammed for anti-tumor therapy using a semi-synthetic organism. *Nat Commun* 12
- Quayle SN, Girgis N, Thapa DR, Merazga Z, Kemp MM, Histed A, Zhao F, Moreta M, Ruthardt P, Hulot S, *et al* (2020) CUE-101, a Novel E7-pHLA-IL2-Fc Fusion Protein, Enhances Tumor Antigen-Specific T-Cell Activation for the Treatment of HPV16-Driven Malignancies. *Clin Cancer Res* 26: 1953–1964
- Sahai E, Astsaturov I, Cukierman E, DeNardo DG, Egeblad M, Evans RM, Fearon D, Greten FR, Hingorani SR, Hunter T, *et al* (2020) A framework for advancing our understanding of cancer-associated fibroblasts. *Nat Rev Cancer* 20: 174–186
- Salmon H, Remark R, Gnjatic S & Merad M (2019) Host tissue determinants of tumour immunity. *Nat Rev Cancer* 19: 215–227
- Seoane J & Gomis RR (2017) TGF- β Family Signaling in Tumor Suppression and Cancer Progression. *Cold Spring Harb Perspect Biol* 9: 22277–22278
- Sordo-Bahamonde C, Lorenzo-Herrero S, Payer ÁR, Gonzalez S & López-Soto A (2020) Mechanisms of Apoptosis Resistance to NK Cell-Mediated Cytotoxicity in Cancer. *Int J Mol Sci* 21
- Stanton SE & Disis ML (2016) Clinical significance of tumor-infiltrating lymphocytes in breast cancer. *J Immunother Cancer* 4: 59
- Triulzi T, Cecco L De, Sandri M, Prat A, Giussani M, Paolini B, Carcangiu ML, Canevari S, Bottini A, Balsari A, *et al* (2015) Whole-transcriptome analysis links trastuzumab sensitivity of breast tumors to both HER2 dependence and immune cell infiltration. *Oncotarget* 6: 28173–28182
- Wu F, Yang J, Liu J, Wang Y, Mu J, Zeng Q, Deng S & Zhou H (2021) Signaling pathways in cancer-associated fibroblasts and targeted therapy for cancer. *Signal Transduct Target Ther* 2021 61 6: 1–35
- Yoshihara K, Shahmoradgoli M, Martínez E, Vegesna R, Kim H, Torres-Garcia W, Treviño V, Shen H, Laird PW, Levine DA, *et al* (2013) Inferring tumour purity and stromal and immune cell admixture from expression data. *Nat Commun* 4: 2612
- Zamarron BF & Chen W (2011) Dual roles of immune cells and their factors in cancer

development and progression. *Int J Biol Sci* 7: 651–658

Reviewers' Comments:

Reviewer #1:

Remarks to the Author:

Comments to Authors

This manuscript has undergone a major revision that has contributed greatly to its improvement. One of the major points that has been raised in the original review – namely the identity of the immune cells that take part in the process and their mode of action – was only partly addressed in the revised version and will need to be supported by additional data. More details are provided below. In addition, few other points require the attention of the authors.

Major points:

1. The identity of the immune cells that take part in the process, and their mode of action:

In the original review, the potential roles of T cells in the process, and of non-ADCC mechanisms were extensively discussed.

To accommodate with these comments, the authors have added Supplementary Figure 5e, in which they demonstrate that antibodies to CD16 (Fc receptors) prevented the process, thus supporting an ADCC-mediated mechanism. However, the antibodies inhibited the response not only in the presence of FAP-IL2v, but also when control DP47-IL2v was used. Moreover, this analysis was performed in a setting established with NK cells and not with total preparation of immune cells (that includes T cells), therefore not excluding the possibility that T cells are also involved in the process.

Also, the authors demonstrate that the addition of NK cells to the ex vivo cultures does the work (Fig. 5g). However, the effect seems to be milder than the one obtained in Figure 4g with immune cells (that contain also T cells), using the same tumor cells. Therefore, the possibility that T cells are involved in the process is still valid.

To address these points, the authors need to repeat the experiments of Figure 4g – which is the original setting they established – when immune cells are used and not NK cells, in the presence of neutralizing antibodies to CD16, and in parallel with neutralizing antibodies to CD3 (or CD4, CD8). Due to their very high relevance to the study, it is recommended to include these data in the main body of the manuscript, and not in supplementary figures.

2. GSEA datasets

The GSE50948 dataset includes not only HER2+ breast cancer patients, but also patients diagnosed with inflammatory breast cancer. These are two very different types of breast cancer.

The authors do not indicate what is the number of patients with inflammatory breast cancer in the dataset. If their number is relatively small, they can be omitted from the analysis. If not, the entire GSE50948 dataset will have to be omitted from the study.

Additional points:

1. The authors demonstrated data about IL2RB. Please indicate that NK cells do not express IL2RA, which is the actual IL-2 receptor subunit that is required for high affinity binding of the IL-2 by T cells.

2. Please indicate the number of patients included in each subgroup of each of the two GSEA

datasets (GSE50948: RD, pCR; GSE55348: Rel, RF).

3. Figure 3e: Does the Y axis represent percentages of CD56+ cells? please indicate.

4. Figure 4e: A control without fibroblasts is missing.

5. The use of FAP-IL2v: The idea of directing a FAP-targeting antibody needs to be explained (retention in the tumor).

6. Please note that the PDF file of the Supplementary Tables is not intact; it does not include all tables.

Reviewer #2:

Remarks to the Author:

the authors have addressed my comments

Reviewer #4:

Remarks to the Author:

Thank you for your responses to the issues raised by this review. You have done an exceptional job at addressing the critiques by developing new data or clarifying areas of confusion.

Response to reviewers

We thank the reviewers for their positive evaluation of this revised manuscript. You'll find below a point-by-point reply detailing how we have addressed reviewer 1 remaining concerns. We have labeled new data in blue in the text.

Major points:

1. The identity of the immune cells that take part in the process, and their mode of action: In the original review, the potential roles of T cells in the process, and of non-ADCC mechanisms were extensively discussed. To accommodate with these comments, the authors have added Supplementary Figure 5e, in which they demonstrate that antibodies to CD16 (Fc receptors) prevented the process, thus supporting an ADCC-mediated mechanism. However, the antibodies inhibited the response not only in the presence of FAP-IL2v, but also when control DP47-IL2v was used. Moreover, this analysis was performed in a setting established with NK cells and not with total preparation of immune cells (that includes T cells), therefore not excluding the possibility that T cells are also involved in the process. Also, the authors demonstrate that the addition of NK cells to the ex vivo cultures does the work (Fig. 5g). However, the effect seems to be milder than the one obtained in Figure 4g with immune cells (that contain also T cells), using the same tumor cells. Therefore, the possibility that T cells are involved in the process is still valid.

This referee indicates that the effect of FAP-IL2v/trastuzumab (TTZ) in the presence of NK cells alone (displayed now in figure 5g and supplementary figure 5d) may seem to be milder compared to the one obtained with ICs in figure 4g. However, a similar minor reduction in treatment efficacy is observed with ICs in figure 6i. In this case, the effect of FAP-IL2v/TTZ in the presence of NK cells (figure 5g) is comparable to the one obtained with ICs (that contain also T cells) in figure 6i.

In order to clarify this point, we provide here below a comparative study of the pooled data displayed in figures 4g, 5g, 6i and supplementary figure 5d for FAP-IL2v/TTZ efficacy against the same cancer cells in the presence of either ICs or purified NK cells.

These data indicate that no significant overall difference was observed in FAP-IL2v/TTZ anti-cancer efficacy in the presence of either ICs or NK cells alone.

Collectively, we agree that slight variations can occur between sets of experiments. This may be explained by the heterogeneity between candidates that donated ICs over the course of this study. Notwithstanding the variability introduced by this factor, a similar effect of FAP-IL2v/TTZ in improving anti-cancer response was repeatedly observed in presence of whole ICs preparation or when inoculating NK cells alone. Of note, CD16 constitutes the main receptor triggering ADCC. We used CD16 blocking antibodies to further conclude that FAP-IL2v is potentiating trastuzumab-induced ADCC by NK cells.

To address these points, the authors need to repeat the experiments of Figure 4g – which is the original setting they established – when immune cells are used and not NK cells, in the presence of neutralizing antibodies to CD16, and in parallel with neutralizing antibodies to CD3 (or CD4, CD8). Due to their very high relevance to the study, it is recommended to include these data in the main body of the manuscript, and not in supplementary figures.

As described and discussed previously in this manuscript, NK cells robustly infiltrate/expand in our model upon dual FAP-IL2v/TTZ treatment, as opposed to other immune cell subtypes such as T cells, which remain unchanged. Since infiltration/expansion of immune cells is indicative of cytotoxic responses triggered by immunotherapy, the lack of T cell infiltration/expansion upon treatment reported in our model would suggest a minor role in the observed responses.

Consequently, we decided to focus on assessing the ADCC-mediated effects of NK cells. As stated above and in this manuscript, NK cells are recapitulating FAP-IL2v/TTZ anti-cancer response through ADCC mechanism. Although we agree on the fact that many other immune cell subpopulations might be involved during anti-tumor responses, we believe that answering this question would be a subject for a broader study.

Manuscript was modified accordingly (line 264 to 280).

The whole immune cells preparation used in HER2+3DiBC contains different cell subtypes that may impact anti-cancer response. Yet to take place, anti-cancer immunity needs immune cells to infiltrate into the tumor. We thus aimed to establish the cellular composition of the immune infiltrate that associates with response to treatment...

In line with this reviewer's suggestion, figure 5g (previously labeled supplementary figure 5e) shows now the superior anti-cancer activity of NK cells upon FAP-IL2v/TTZ regimen and its inhibition in presence of anti-CD16 neutralizing antibodies.

2. GSEA datasets

The GSE50948 dataset includes not only HER2+ breast cancer patients, but also patients diagnosed with inflammatory breast cancer. These are two very different types of breast cancer. The authors do not indicate what is the number of patients with inflammatory breast cancer in the dataset. If their number is relatively small, they can be omitted from the analysis. If not, the entire GSE50948 dataset will have to be omitted from the study.

GSE50948 subset that was used for this study includes locally advanced and inflammatory HER2+ BC patients treated with trastuzumab.

Following this reviewer's request, we have now excluded the women diagnosed with HER2+ inflammatory breast cancer from this study. ESTIMATE, GSEA and correlation analyses are giving results overall similar to the ones obtained with the cohort that included locally advanced and inflammatory HER2+ BC. These data are now displayed in this manuscript.

Additional points:

1. The authors demonstrated data about IL2RB. Please indicate that NK cells do not express IL2RA, which is the actual IL-2 receptor subunit that is required for high affinity binding of the IL-2 by T cells.

We assessed IL2RB expression in immune cells compared to cancer cells and fibroblasts to determine which compartment might respond to IL2 in our *ex vivo* model. Our data indicate that immune cells are expressing IL2RB. In contrast, cancer cells and fibroblasts do not show any detectable expression of this receptor, thus indicating that ICs are the main responders to IL2 in HER2+3DiBC (supplementary figure 2c). We neither investigate nor discuss the particular expression of IL2RA/B by NK cells.

Corresponding sentence was modified to further clarify this point:

Line 184 to 186: ... *IL2 receptor B (IL2RB) expression was only detectable in ICs compared to BC cells and BFs (supplementary figure 2c)*...

2. Please indicate the number of patients included in each subgroup of each of the two GSEA datasets (GSE50948: RD, pCR; GSE55348: Rel, RF).

Patient numbers in RD, pCR, Rel and RF subgroups are now indicated in Figure 1 and in Material and Methods (*Generation of gene expression signatures and association with clinical parameters* section).

3. Figure 3e: Does the Y axis represent percentages of CD56+ cells? please indicate.

Figure 3e displays the number of CD56+ cells detected in HER2+ tumor cores.

Figure legend was modified accordingly:

e) *CD56+ cell number detected in HER2+ BC tumor cores comparing relapse-free (blue) to relapsing patients (red)*...

4. Figure 4e: A control without fibroblasts is missing.

Control FAP immunostaining for Figure 4e was performed on cultured cancer cells (HCC1954). A representative image was incorporated as supplementary figure 3b. As previously shown in figure 4b and supplementary figure 5c, FAP protein expression is not detected in cancer cells.

5. The use of FAP-IL2v: The idea of directing a FAP-targeting antibody needs to be explained (retention in the tumor).

In agreement with this referee's comment, the manuscript was modified as follows:

Line 203 to 208: *...severe adverse reactions are associated with systemic IL2 treatment...^{33,34}. Consequently, an original immunocytokine composed of IL2v...fused to a FAP-targeting antibody...was designed to direct and retain IL2 activity into FAP-expressing TME while reducing IL2 adverse systemic effects³⁵.*

Line 222 to 226: *...Collectively, our findings indicate that HER2+ BC tumors resistant to anti-HER2 mAbs display upregulated CAF-S1/pCAF/FAP expression as well as reduced immune cells abundance and low IL2 activity. Therefore, the capacity of FAP-IL2v to increase IL2 bioavailability into FAP-enriched stroma may strongly benefit those patients unresponsive to trastuzumab.*

6. Please note that the PDF file of the Supplementary Tables is not intact; it does not include all tables.

This issue may be inherent to the formatting process occurring when uploading files in the tracking system.

Reviewers' Comments:

Reviewer #1:

None